# Inka *Unku*: Imperial or provincial? State-local relations

**Jacqueline Correa-Lau**[1]*, **Carolina Agüero**[2], **Jeffrey Splitstoser**[3], **Ester Echenique**[4], **Tracy Martens**[5], **Calogero M. Santoro**[1]*

**1** Instituto de Alta Investigación, Universidad de Tarapacá, Arica, Chile, **2** Sociedad Chilena de Arqueología, Santiago, Chile, **3** Department of Anthropology, The George Washington University, Washington, D.C., United States of America, **4** Departamento de Antropología, Universidad de Tarapacá, Arica, Chile, **5** Department of Archaeology and History, La Trobe University, Melbourne, Australia

* jcorreal@academicos.uta.cl (JCL); calogero_santoro@yahoo.com (CMS)

**Data Availability Statement:** All relevant data are within the paper and its Supporting Information files.

**Funding:** ECOS200056, Sra. Jacqueline Correa-Lau (CMS, JC-L) https://www.anid.cl/redes-estrategia-

## Abstract

Standardized Inka tunics, or *unku*, were created under the auspices of the state as symbolic expressions of its expansionist power. To ensure these textiles acquired the status of effective insignias of power and territorial control, the Inka established and imposed technical and stylistic canons for their production (*techne*) by means of highly-skilled state weavers. In the provinces, social groups that came under imperial rule, local expert weaving agents adopted the conventions of the state and included meaningful symbolic elements of the idiosyncrasies, traditions, and experiential knowledge of the local community (*metis*). We therefore propose that this was not a unidirectional process and that the Caleta Vitor Inka *unku* (hereon referred to as the CV *unku)*, presented here, reflects a syncretism promoted by local weavers. In terms of methods, we have developed a decoding tool for the *unku*, with the aim of distinguishing state from local hallmarks, thereby revealing the syncretic complexity of these iconic tunics. This methodological tool is based on a series of standard analytical parameters and attributes linked to morphological, technological, and stylistic features, which we applied to the CV *unku*. Unlike others, this *unku* does come from a looted tomb but was scientifically excavated in a cemetery located in the Caleta Vitor Bay in northern Chile. By deconstructing the CV *unku* we determined the steps in the *chaîne opératoire* at which local technical and stylistic elements were incorporated, thus affecting or transforming, in part, its emblematic imperial imagery. This study also marks a step forward in our understanding of a syncretic landscape that combines the state worldview and organized production system (imperial Inka) with craft-production practices that were rooted in provincial and local communities (provincial Inka).

## Introduction

Over the course of their territorial expansion, the Inka incorporated several social groups, each with its own traditional way of producing material culture, imposing upon them a set of state canons that were systematically reproduced throughout the empire. However, given the

y-conocimiento/programa-de-cooperacion-
cientifica-ecos-anid/ FONDECYT 1201687, Sr.
Calogero M. Santoro (CMS) https://www.anid.cl/
proyectos-de-investigacion/fondecyt-regular/
NCN19_153, Sr. Calogero M. Santoro (CMS)
https://www.anid.cl/centros-e-investigacion-
asociativa/nucleos-milenio/ UTA Mayor 3759-22,
Sra. Ester Echenique (EE) https://www.uta.cl/index.
php/concurso-de-proyectos-mayores-de-
investigacion-cientifica-y-tecnologica-utamayor-
2022/ The funders had no role in study design,
data collection and analysis, decision to publish, or
preparation of the manuscript.

**Competing interests:** The authors have declared
that no competing interests exist

capacity of weaving agents to apply their own thought processes and expertise to their craft, we believe that, as well as appropriating and reproducing the state's morphological, technological, and stylistic canons, they also imbued fabrics with attributes from their own cultural traditions. The result was textiles with visible adherence to state canons, while local hallmarks were ingrained in less obvious technical, formal, and aesthetic features. The combination of techniques and styles both reflected and shaped the syncretic relationship between the state (imperial Inka) and the provincial or local communities (provincial Inka).

In this regard, we address the importance of the relationship between state and local knowledge, which brought agents into a dialectical arena in which state ideologies could be negotiated. The importance lies in the fact that, despite the scale of the social and military structure of the Inka State, its expansive, economic and ideological success depended to a large extent on the agreements reached, peacefully or belligerently, with the local communities of the provinces integrated into the empire. Among the dialectical scenarios was the textile industry, in whose production the technical and idiosyncratic canons of the state were negotiated with those of the communities themselves. In this context, the *unku*, as symbols of imperial power, are key elements for materially visualizing the dialectical relationship between the state and the local communities.

For the analysis of this dialectical arena, we use the concepts of *metis* and *techne* [1] to facilitate an understanding of the dialectical relationship between these agents. On the one hand, *metis* refers to knowledge acquired through experience. This type of knowledge is generated locally and as a result is tied to the place of origin. *Techne*, on the other hand, is knowledge taught or transmitted as a formal and established discipline. It differs radically from *metis* in regard to organization and transmission by being codified, taught, and modifiable [1:319]. Unlike the particular and contextual *metis*, *techne* combines discipline and technical knowledge, enabling the Inka State to convey ideological parameters that were reproducible in textiles and other emblematic goods throughout Tawantinsuyu.

The concepts of *metis* and *techne*, or the relationship between local and state knowledge, can be materialized and understood through both high-visibility and low-visibility attributes, commonly used in the archaeological literature of enculturation. High-visibility attributes are intentional signaling features such as decorative/stylistic designs that denote standardized choices made by the government to transmit and create a sense of being part of larger corporate polity. Low-visibility attributes are technological features, such as raw materials, yarn types, or a particular textile structure, linked to group identity and community.

Costin [2] makes reference to the fact that studies of Inka textiles have focused on their meaning from the perspective of consumption rather than the circumstances of their production. This, despite almost all imperial subjects being engaged, to a greater or lesser extent, in textile production, whether procuring or processing raw materials, spinning fiber, weaving fabric, or finishing a garment [3, 4]. We estimate that throughout this *chaîne opératoire*, the state imposed its own set of canons with the aim of standardizing the production process. Hence, the kind of syncretism proposed here derived from the skills of local weavers in adhering to the state's mode of "doing", while adding their own traditional imprint, visible or otherwise, local or personal.

The technological styles and symbolic structures of the textiles produced for the state constituted an active tool of social action [5] insofar as their production depended on the weavers of the local communities who had to follow the state model. In this active role of perpetuating state canons, local weavers had the insight to introduce subtle technical and stylistic changes throughout Tawantinsuyu. This is reflected in the degree of variability that can be observed in *unku*, like the Caleta Vitor Inka *unku* (hereon referred to as the CV *unku*). A previous study of the CV *unku*, conducted by Martens et al. [6], confirmed that it was an authentic imperial Inka

garment, as it shared key imperial technical and stylistic characteristics—observed in 36 examples of *unku* from different museums across Europe and the United States of America—that were compiled for her comparative study and recorded in a database [6]. In the process of extending the study of this particular *unku* we realized that state parameters were accompanied by features that did not conform to the Inka style as defined in previous classifications [7].

To further analyze the *unku*, while following our theoretical perspective, we developed a methodological tool—based on parameters and attributes that consider morphological, technological, and stylistic features—that bridges the knowledge gap between high-visibility and low-visibility attributes, where morphology relates to textile composition. Technology links social productive activities and their interactions with material culture [8], which means that "technology, as embodied material practice, is a socially charged and materially grounded arena in which agents express and negotiate social relationships, establish and express value systems, and give meaning to the object world" [9:162]. Technology, then, is recognized as a meaningful social activity, driven through agency processes [10]. Style here refers to "a form of non-verbal communication, through doing something in a certain way that conveys information about relative identity" [11:8]. Our methodological tool is aimed to differentiate where in the *chaîne opératoire* the state canons differ from the local ones, thereby revealing the syncretic complexity of these iconic tunics (cf., Mary Frame's [12] idea that hybridization in textile design was an effect of the technology used during the manufacturing process). It is designed for in-depth physical analysis to infer information about the identity of the intended wearer of the garment, as well as to understand its practical and functional purpose [13]. This is based on the hypothesis that technical and/or stylistic differences in the *unku* might represent differences in the status of the wearer, which include the ruling elite, administrators, priests, and members of the military. Because each *unku* was woven for one of these characters, each with a specific role and status, were weavers allowed or did they take a significant control over the way they conformed to (or deviated from) imperial versus local weaving styles and parameters? Furthermore, in some cases, *unku* acquired additional elements and underwent modifications for reuse and when being discarded.

Historical references describe with great precision the characteristics of the garments worn by the Inka, his administrative entourage, and by members of the army. Previous studies have focused on the visible, imperial, technical, and stylistic features, which is inconsistent with the Inka strategy that respected local cultural practices and the resulting hybridization of material culture, as we propose was the case of local *unku*. Local individuals or groups negotiated their relations with the social structures of the state, setting in motion a dynamic social agency, which played an essential role in shaping technology [9]. This may have entailed shifts in the relationship between status and basic ideological concepts among interacting social agents. Our analysis of this process enables us to visualize the relationship between technological action and the propagation of state power and ideology.

## Inka *unku*, imperial material imagery

The Inka *unku* was a male garment made by expert weavers [14], whose knowledge, techniques, and skills were both valued and endorsed by the state which implemented the textile programs, as reflected in their high degree of standardization. Paramount homogenizing political, economic, and social organizations, such as the Inka State, established socially recognized morphological, technological, and stylistic elements that had to be considered by weavers. These specialists could not disregard the fact that their products were intended to convey power, prestige, status, or serve as the state's territorial insignia, entrusted to dignitaries from within the core society and beyond [15, 16]. An example of textiles carrying this semantic load

can be seen in the context of the mountain-top female Inka burials of Cerro Esmeralda on the coast of the Atacama Desert, northern Chile, which would have formed part of the Inka expansion along the coast and to the south of the imperial center in Cusco. On the slopes of this mountain that rises steeply above the Pacific Ocean near the Inka-controlled Huantajalla silver mine, a cache of textiles and other objects were found that accompanied the burial of two young women [17:36, 18–21]. Beyond their intrinsic value, these textiles have a profound symbolic meaning tied to the legitimation of Tawantinsuyu expansion and domination [7, 22–34]. Similar meanings are also attributed to the *cumpi* (finely woven cloth), often adorned with rectangular geometric motifs or *tocapu*. Highly prized for their symbolic value, this attribute indicated noble status [13:175–176].

*Unku* were originally produced for daily use by the ruling classes [35, 36]. As such, they were considered imperial attire, not to be worn by anyone unless gifted by a government official [7] or provided for a special purpose such as exercising a military role [15, 37]. Some sumptuary goods were also given as part of the *mitimae* program of forced resettlement. The wealth of the Inka was measured in part by their "generosity" in sharing used cloth among the leaders of the different ethnic groups that were absorbed into the state. Those gifted with such garments would be obliged to wear them for special civic-ritual ceremonies. The garments would also be awarded on an annual basis to soldiers who had displayed valor in battle. The clothes appeared to be new, but on some occasions, repaired items were also distributed [14]. In Andean communities there is a relationship between clothing and people, insofar as the possession of certain types of clothing confers special social and cultural status to the wearer [22, 38]. Unused pieces were presented as sacrificial offerings or used for dressing the dead, reflecting an intimate association between textiles and death [14:157]. We have interpreted this relationship as reinforcing the social, ethnic, cultural, and gender roles of the individuals buried with this type of clothing, maintaining a sense of a community through textile production, in which social boundaries were strengthened through exchange networks, similar perhaps to the practices proposed by Ann Peters for the Paracas Necropolis [4]. At the same time, these garments represented imperial power for provincial leaders or officials in their natural setting, deriving from deliberate political action by the state to distribute these garments as part of political, economic, and ideological agreements.

The state imposed a series of labor obligations and commitments through *mit'a* (asymmetric reciprocal labor system) using different coercive measures, including political, economic, military and even matrimonial agreements with local communities. In this system of reciprocal relations, the state provided the necessary resources (raw materials, tools, architectural facilities, etc.) for the performance of assigned tasks [14, 39]. The *mit'a* agreements also implied great acts of generosity on the part of the state as another way of committing local communities to participate in this reciprocal process [14]. In this context of generosity, we believe that the state was also able to transmit the stylistic, architectural, morphological, and technological patterns to be applied in local production, all of which involved acculturation processes [40, 41].

Standardization, such as that found in the *unku*, is likely related to the need of the state to reproduce its symbols of power. Therefore, garments had to adhere as strictly as possible to the standard parameters that gave these pieces an eminently imperial character; however, it has also been pointed out that one of the political strategies of the Inka state was to coordinate the diversity of the peoples, with their different cultures, languages, and traditions, without trying to homogenize them completely. The Inka promoted diversity and allowed local groups to produce material culture and conduct local ritual behavior in ways that represented their ethnic identity [14, 42]. In short, the relationship between the state and local communities was a dialectical process that varied from region to region and village to village, depending on the

pressure exerted by the state and the receptivity of local groups [40, 43]. In our opinion, these historical contexts would have generated the conditions for local weavers to introduce morphological, technological, and stylistic features with low visibility, attributes which did not radically transform the imperial character of these iconic objects that represented the state.

To identify the state-imposed aesthetic in the production of the CV *unku*, we looked at the typologies proposed by John Rowe [7] and Ann Pollard Rowe [30] for Inka tapestry tunics, which consist of six styles. Ann Pollard Rowe expanded on John Rowe's findings to develop a system that includes formal, stylistic, and structural characteristics of *unku*, both imperial and provincial [29, 30, 44], concluding that Inka *unku* can usually be identified by the following attributes: construction (woven with the warp in the short direction), woven-in neck slot (as opposed to cut), embroidered edges (including armholes and neck slot), dimensions being longer than wide, and the use of weft-faced plain weave with discontinuous, simple-interlocked, two-ply, camelid fiber wefts interweaving three-ply warps (cotton or camelid fiber).

These typologies have been used by many researchers to broaden the discussion of the symbolic, political, and social meanings both embedded and visible in *unku* [15, 16, 32, 45–47]. All the tunics "are of Inka size and proportions, which appear to be the key factors in what was appropriate for Inka men to have" [30:33]. For example, following John Rowe's typology, Pillsbury [15, 16] focused her analysis on the *unku*'s spatial composition. Discontinuous warps and wefts were presumably woven "to signal the office of the person who wore that class of garments" [48:230]. Consequently, Dransart has suggested that "it is possible to see different levels of social articulation in these garments" [48:230]. The interlocking tapestry to create binary oppositions and the use of space were identified as a dominant textile features among altiplano and highland social groups in the southern Andes [49:116]. Katterman [25] sought to draw a distinction between different social ranks by focusing her technical analysis on finishes. Bray [50] and Frame [51] presented a perspective based on design and iconography. Several authors, including Vreeland [52], Vanstan [53], and Desrosiers [54, 55], analyzed the techniques employed in making *unku*, as well as the use of looms and weaving types.

Most of the studies of *unku* noted above approach them from a technical and stylistic perspective primarily focused on identifying Inka influence on the provinces, which we consider a unidirectional perspective, concerned mainly with high-visibility attributes, or *techne*. In contrast, most of the 55 *unku* that were analyzed for the present study—from several museum collections from around the world (S1 Table)—were studied from the perspectives of both *techne* and *metis*. In this paper, we demonstrate that, when creating local *unku*, expert weavers followed the state canons—morphological, technological, and stylistic—to reproduce the emblematic symbols embedded in these imperial garments (*techne*). Simultaneously, they integrated meaningful symbolic elements representative of local idiosyncrasies, traditions, and experiential knowledge (*metis*). Consequently, the emblematic symbol of *unku*, instrumental for the expansion of the state, would have resulted from dynamic relationships between the Inka and the local community where expert weavers played a key role as agents of change.

Although local weavers would have been required to adhere to strict state canons when crafting *unku* and other textiles, as has been suggested for *chuspas* (coca bags) [27], they likely would have also found a way to develop and incorporate elements of local style, depending on their own life experiences and social milieu at the time of weaving. These changes in the technical, formal, and aesthetic aspects of these textiles, resulting from the interaction between the state and the local community which we believe represent the concepts of *techne* and *metis*, are part of what is known as provincial Inka [25, 29, 56, 57]. Yet we also suggest that certain stylistic and/or technical hallmarks introduced by local weavers were intentionally made less obvious or evident so as to remain in keeping with state conventions. This, given that the functional purpose of the garments and the essentially imperial values associated with them,

was reinforced by means of adherence to the state-imposed canons as visible expressions of power, prestige, and ideology.

In the Atacama region [58], local weavers introduced a series of hallmarks to the imperial attributes of the *unku*. For example, multiple wefts were introduced, a traditional textile technique that became a concealed technical trait found in textiles during the Inka period [59–61]. On the coast of Arica, the continued use of single wefts has been recognized [60, 62]. Other examples of this type of syncretism have been described for ceramic production in the area inhabited by the Diaguita people. For instance, González [63] suggests that specialized *mitmakuna* (Inka State forced settlers) would have been brought to this province to produce, and ultimately teach, locals how to reproduce non-variant imperial ceramics. Interactions between these artisans would have led to the merging of local technological procedures and stylistic expressions with those imported from Cusco, giving shape to a provincial Inka tradition. Williams et al. [64] have shown that local ceramic artisans tried to emulate, as much as possible, the appearance of imperial styles, and in that process added certain traditional procedures, such as the use of local raw materials that differed from those used in the state centers of ceramic production (*ollero* villages) in the Andean zone [65]. Their ceramic analysis demonstrated that, although the raw materials were different, the artisans tried to give the vessels an imperial appearance (Hughes [27] points out a similar situation for the production of *chuspas*). This symbiosis combined technical, iconographic, and morphological characteristics whilst leaving space for local modifications. Provincial Inka styles were generated in this way, produced by local artisans who set about reproducing styles that appeared to be like those emanating from Cusco. Cusqueño-like vessels were produced throughout the South Central Andean region, including the Diaguita area and in Northwest Argentina [66], and we can recognize a broad variety of styles ranging from goods with typical Inka hallmarks to objects inspired primarily in accordance with local traditions [49–56].

While visibly reflecting the state's aesthetic canons, provincial Inka styles were characterized by additional features or modifications, such as the use of different color combinations, geometric designs, or some form of iconography associated with the local landscape [67]. It appears, therefore, that the state posed no impediment to the application of local canons during the production process. To the contrary, it is likely the state allowed flexibility for agents to employ certain local practices that could be reproduced by hand while incorporating stylistic expressions of the state [14].

To test the hypothesis that the CV *unku* was produced by specialized local weavers who incorporated their own symbolic and meaningful elements, our case study focuses on the *unku* found in Caleta Vitor Bay, northern Chile [6]. The CV *unku* is an exceptional example, not only for that area but also for the entire Andean region. Unlike most of the rare examples of *unku*, the CV *unku* (Fig 1) comes from the archaeologically excavated remains of a previously looted funerary context comprising a series of elements that accompanied the burial. Within this assemblage, the CV *unku* is the most eloquent form of material expression related to the Inka; therefore, we suggest that this textile was possibly locally produced in the context of a work commitment imposed under the *mit'a* system [14].

## State and provincial *unku*: Morphology, technology, and style

In the earliest phases of Andean history, vegetal fibers such as *Asclepias* sp. and *Cyperaceae* sp. were commonly used to make fabrics [68–70]. Cotton does not appear on the coast and valleys of northern and central Peru until the Middle Archaic, becoming popular by the Late Preceramic period (5000 cal yr BP), largely supplanting the use of vegetal fiber in fabrics [62, 69, 71, 72]. Its use continued until the Late Horizon or Inka period. Camelid fiber does not appear on

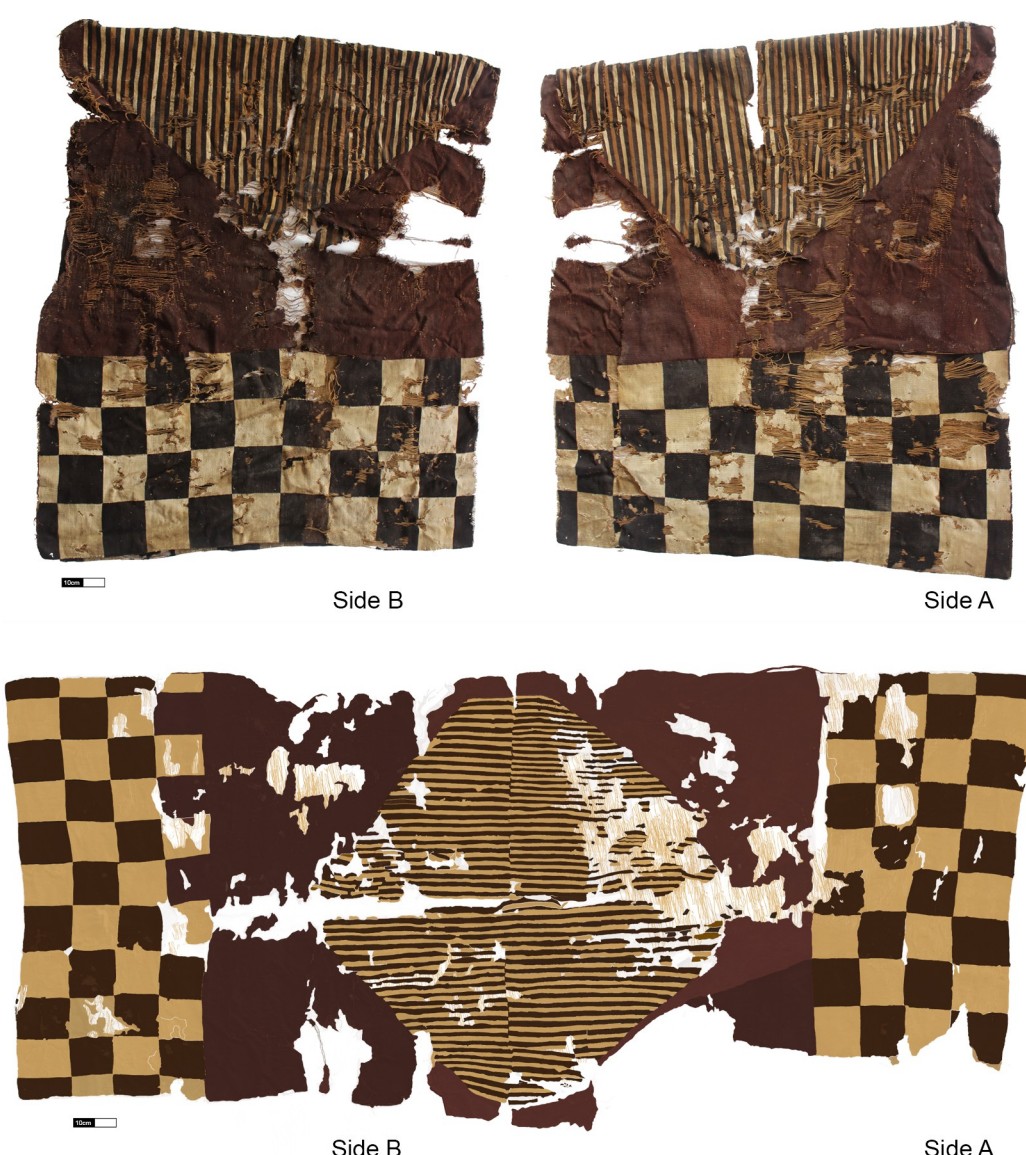

**Fig 1. *Unku* found in Caleta Vitor Bay.** Top: sides A and B from the wearer's point of view (photographs by Paola Salgado); bottom: illustration of the tapestry tunic from the weaver's position and viewpoint (illustration by Paola Salgado).

the north coast until the Early Intermediate Period [73]. In contrast, in the valleys and coast of southern Peru and northern Chile, vegetal fiber was followed by camelid fiber, and cotton was introduced during the early Formative (ca. 4000 cal yr BP) [68, 72, 74]. In Caleta Vitor Bay, Martens and Cameron [74] confirmed that fabrics made of vegetal fiber were common since the Early Archaic (ca. 9000–8000 cal yr BP). By the Late Formative, however, this raw material drops off as camelid fiber becomes the most common material. Similarly, the use of cotton was very limited, but became overwhelmingly dominant up to the Inka period [74].

Tapestry warps were traditionally made of white cotton on the coast and valleys of central and northern Peru during the Inka period (Late Horizon). While camelid fiber was less commonly used for warps in tapestry, it was the dominant fiber used for wefts [29]. Similarly, it has been noted that Inka-period textiles from Tarapacá Viejo (the valley administrative center)

in northern Chile were also woven in weft face with white cotton warps and camelid fiber wefts [31, 75]. In contrast, at Caleta Vitor Bay, all *chuspas* have camelid fiber warp and weft. Cotton was almost absent.

Cotton textiles from Huaca Prieta on Peru's north coast (near the modern city of Trujillo) show that spinning practices used to produce yarns with an "S" twist were well established over 6000 years ago [62]. On the south coast of Peru, cotton was usually spun with a "Z" twist since at least ca. 3000 cal yr BP. These early textiles throughout coastal Peru, however, often included "counter-spun" yarns and yarns that were plied with Z- and S-twists, leading to the creation of new yarn parameters [62, 76]. In pre-Hispanic times, final twist had symbolic connotations for ritual or everyday activities in the Cusco region [77–79]. Arnold [80] also noted that spin and ply direction continues to have a sociocultural and communicative properties. The predominant form is S(2z) (two Z-spun yarns plied in the S-direction), although in exceptional cases, such as for ritual use, both twists are combined. In some fabrics, the combination of S and Z final twist, where yarns with a Z twist are called *lloque* (left handed) yarns was a visual device that derives from aesthetic considerations [80:99]. Both Rowe and Cohen [81] and Phipps [82] note its presence, frequently inserted in the edges of fabrics made in the highlands from Colonial to modern times, where it served multiple functions that ranged from functional—keeping edges from rolling—to aesthetic imparting a herringbone effect. Splitstoser [83] found counter-spun yarns combined with yarns with regular twist, forming repetitive patterns in archaeological textiles from both Cerrillos and Huaca Prieta [62, 83]. Their function is unknown, however, because unlike most highland textiles, which are made of camelid fiber, the archaeological textiles studied by Splitstoser [83] are made primarily of cotton and counter-spun yarns that are found throughout the textiles, not just primarily in the selvages.

The degree of control the Inka exacted over the production of *unku* garments is also evident in the procedures put in place to ensure the amount of labor required to make each textile was consistent. For instance, Costin [2] notes that the state introduced a quadrangular area as standard for creating large garments. Ann Pollard Rowe [44] and John Rowe [7] documented at least six examples where this metric is evident, suggesting standardized production and a degree of supervisory control.

Large textiles were woven using special looms, propped against a wall, with a vertical structure that consisted of cross bars, top and bottom, as well as heddles. This description by John Rowe [7] is based on the drawings of Guamán Poma [84]. Others, such as Rojas and Hoces de la Guardia [85], Oakland [86], and Conklin [87], have pointed out that broader pieces may have been worked simultaneously by more than one person. Costin [2] and John Rowe [7] believed this interlocking tapestry technique to have been a state technology. Unlike other Andean techniques, both their structure and designs were part of an inseparable system [2, 5]. The interlocking tapestry technique [88] produces designs that entirely conceal the warp, generating abrupt changes of color and shapes that culminate in a complex decorative composition without the restrictive right-angles resulting from many other techniques. Discontinuous warps [87] have been known since the early phases of the Nazca culture [89] and were used for making openings, such as the neck slot in *unku*, which were finished with warp chaining. Heading cords, located at both warps ends of the textile and at the ends of the discontinuous warps, lent structure and support to the piece.

John Rowe [7] identified six Inca tapestry-tunic styles: Black and White (BW), Inka Key (IK), *Tocapu* Waistband (TW), Diamond Waistband (DW), All Over *Tocapu* (AOT), and Zigzag Waistband (ZZW). These styles are defined based on the following components: diamonds or rhomboids; stripes or bands; checkerboard; *tocapu* (geometric motifs enclosed in quadrangles); and zigzags. Diamonds or rhomboids, which appear repeatedly on ceremonial objects and mainly on Inka tunics [7, 50], may be introduced as iconographic elements or form part

of the structure, enclosing spatial areas of the tunic into two or four discrete areas of elaboration. The geometric shape of the diamond or rhomboid design, with its symmetry and concentricity, has been interpreted as a graphic expression of the sociopolitical organization of the Inka State, Tawantinsuyu ("kingdom of the Four Quarters") [50]. The spatial and social correlation generated by the diamond derives from the fact that it can be imaginatively divided into halves and quarters along its symmetrical axes, which together form a visual representation of the Andean world [50]. This motif, called *awaqui* or *auaqui* ("V"-shaped yoke) [37:330, 82:41], is frequently recorded in Inka tunics, but it has precedents in tunics from the Wari culture [90:41, Fig 8D]. *Unku* with *awaqui* were worn by certain rulers and captains from Inka nobility, although this was relaxed to include other officials and members of the state. This same design continued to be used during the colonial phase in the garments of various high-ranking officials, such as mayors and native tribute collectors [90]. The *awaqui* motif continues to be used in the textile traditions of modern Andean villages, including those in northern Chile, but with some variations including straight or stepped edges, outlines of rows of triangles, and different interior fills [50].

*Unku* with plain stripes located around the arm slits would have been worn by lower-level administrative functionaries [54, 86]. In the area of Arica, the mostly thin, plain-striped, natural-colored textiles have been interpreted as a simplified decorative popularization of this Inka feature [22, 91, 92].

The checkerboard pattern, though recorded in the garments of pre-Inka cultures such as the Wari [37, 93], was one of the Inka State's most used and standardized designs for *unku*. The black-and-white checkerboard pattern also appears on miniature garments associated with male figurines used in offerings at high-altitude sites [90], and checkered patterns were often applied to ritual textiles, *aryballos* (storage jars), *queros* (drinking vessels), paintings on funerary architecture, and rock art. Berenguer [37:Fig 1, 93] interprets this design as a military emblem based on its widespread distribution across the South Central Andean region, Northwest Argentina, and Central Chile. López Campeny and Martel [90], on the other hand, point out that garments bearing these designs were used by the Inka and their entourage when departing from Cusco to visit the provinces.

In contrast, *tocapu* designs were worn exclusively by the Inka and members of the royal family [2], whose task of weaving them was entrusted to the *aqllakuna* (chosen women). These specialized weavers were confined, which would have guarded the technical know-how involved in creating *tocapu*, and preventing the transmission of this knowledge to outsiders [2].

The zigzag design was applied to garments worn by rulers, military leaders, provincial governors, and high-ranking administrators. In the colonies, this same design was extended to ordinary people and provincial noblemen [7:242]. The zigzag design is widely depicted by Guamán Poma de Ayala in illustrations of Andean characters dressed in *unku* [7].

The final stage in the production of an *unku* was to remove it from the loom and fold it in half along its vertical axis. The fold served to mark the area of the shoulders (from the wearer's point of view), and the neck slot was deliberately left open at the center of the tunic so it could easily be placed over the head. Both sides were sewn together to a sufficient height to allow the arms to be inserted. Tiwanaku, Wari, and Inka *unku* from the Central Andes feature high-density figure eight stitching applied mainly in the lateral seams. This generated a double column that combined several colors, which gave an additional decorative element to the piece [7, 24]. The tunics also present the zigzag in a way that combined the displacement and density of the fishbone stitch. Zigzag embroidery was typically applied near the bottom selvage of the tunic. In addition to the zigzag, the selvage edge of tunics shows overcast stitch, known since Wari (AD 650–1000) [94]. For Desrosiers [54], tapestry tunics from the coast are characterized by

their reinforced longitudinal selvages, which can be observed in Tiwanaku, Wari, Inka, and probably colonial pieces. The tunics also retain their heading cords that produce the same effect as a reinforced selvage.

At least some *unku* were apparently stored as "heirlooms". A case in point involves a colonial-era *unku* which was carefully preserved in a cedar box and is now housed at the Museo de América in Madrid (inventory number 14.501). Jiménez [35] interprets this heirloom as a symbol of resistance against the imposition of European power, having been passed down through successive generations.

Murra [14] noted that Pedro Pizarro witnessed baskets of old or used clothing, including *unku*, that the emperor would gift to the people, and which the authors interpret as a political and economic strategy by the Inka to demonstrate generosity in the complex negotiations the state had with various provincial leaders. In archaeological sites from Arica dating to the Inka Period, both along the coast and in the valleys, textiles have been found that show clear signs of repair for subsequent reuse [22]. Large areas of repairs often appear on textile pieces used as garments during this period, suggesting excessive use and increasingly impoverished clothing among local communities [22]. We believe this may be related to fact that garments containing important symbolic information were handed down from generation to the next.

## Material and methods

### Caleta Vitor Bay, area of study

The Inka subdivided the Atacama Desert region into three major provinces: Arica, Tarapacá, and Atacama [95]. Complex imperial architectural features can be recognized in these provinces, such as the agricultural canals and terraces essential for maize production, as well as urban facilities containing *ushnu* (ceremonial platforms) (Fig 2). State emissaries negotiated with representatives from each local community to foster economic growth and strengthen regional and interregional ties. Once agreements between state officials and local authorities had been successfully reached, territories within the Atacama Desert were rapidly and effectively integrated within the imperial system [95].

Upstream from the Vitor or Chaca canyon in the Codpa valley, evidence of this integration is found in the form of a pictograph with an Inka checkerboard design. Set on a vertical panel just above a small cave near the Molle Grande archaeological site, the squares are painted red and white and cover an area of 140 cm high x 140 cm wide. The panel is located on the lower section of a rocky wall of the steep gorge about 8 meters above its floor (Fig 3). Berenguer [37] describes this site along with a series of other locations in provinces to the south of Cusco where this checkered pattern is depicted.

Several archaeological sites, such as Caleta Vitor Bay, have been recorded at the mouth of northern Chile's steep canyons or quebradas. The Caleta Vitor Bay archaeological complex is characterized by broad, deep middens formed by the accumulation of food remains and other elements associated with domestic life along the coast. Along the exoreic coast in Chile's extreme north, Caleta Vitor Bay is one of the few places that was ideal for early human settlement and dates to the early Holocene [96, 97]. It has permanent sources of fresh water and a wide variety of coastal marine resources (e.g., mollusks, fish, algae, birds, and mammals). These marine food resources, combined with abundant deposits of seabird guano—valued and exploited by the Inka as fertilizer for intensive farming in the inland valleys and oases [98–100] —and the Huantajalla silver mine, nestled in Iquique's coastal mountain range, were significant factors in Inka expansion into this territory.

The middens at Caleta Vitor contain tomb clusters, either in their interior or in adjoining sectors, some of which have been looted. The tomb containing the CV *unku* was entered into

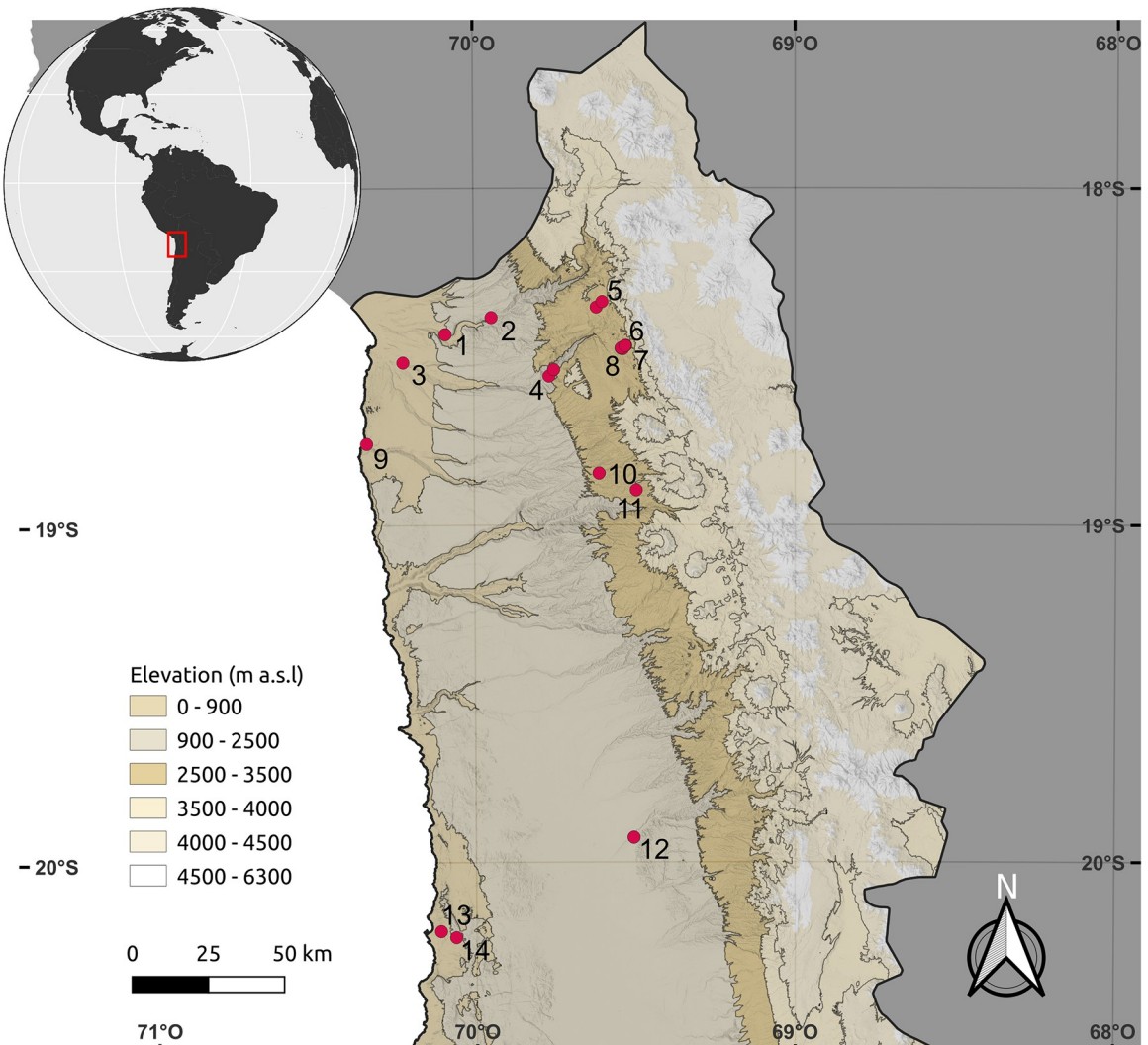

**Fig 2. Distribution of the main sites where imperial Inka elements have been recorded in the provinces of Arica, Tarapacá, and Atacama. (1) Molle Pampa (*ushnu*, *chullpas*); (2) Cruces de Molino (petroglyphs); (3) Azapa 15 (cemetery, *ushnu*, geoglyphs); (4) Pubrisa (village, monumental architecture); (5) Zapahuira complex (*tambo*, *qolqas*, *Qapac Ñan*, *huairas*); (6) Chajpa and Ancopachane architectural compounds; (7) Huaihuarani (hilltop village including plaza, domestic and ceremonial architectures); (8) Molle Grande (pictograph with red and white checkerboard); (9) Caleta Vitor Bay; (10) Inkahullo village (*sunturhuasi* cosmological architecture); (11) Saguara 2 village (domestic and ceremonial architectures: *ushnu*); (12) Tarapacá Viejo village; (13) Cerro Esmeralda (hilltop burial); (14) Huantajalla (silver mine) (map courtesy of Matias Frugone).**

the archaeological record as Tomb N˚1 by Chris Carter [96]. Located on a vertical plane on the eastern edge of zone CV2, the tomb became exposed due to deep and extensive excavations carried out by the Chilean Navy in 1960. In addition to the *unku*, the tomb contained a bow, a clutch of six arrows positioned vertically beside the funerary bundle (Fig 4A), four camelid fiber bags (three square-shaped *chuspa*s and one rectangular *talega*, or domestic bag) (Fig 4B), a leather breastplate, a vegetal fiber mat, a vegetal fiber *capacho* (backpack), and two truncated cone hats with twisted chinstraps and stepped geometrical designs made from a flat, rigid, vegetal fiber base interworked with camelid fiber (Fig 4C) [96:193]. The arrows, made entirely of wood, consist of two parts: the shaft (average length of 43.7 cm), which is stained with red ochre except at the proximal segment where the feathers are attached; and the arrowhead,

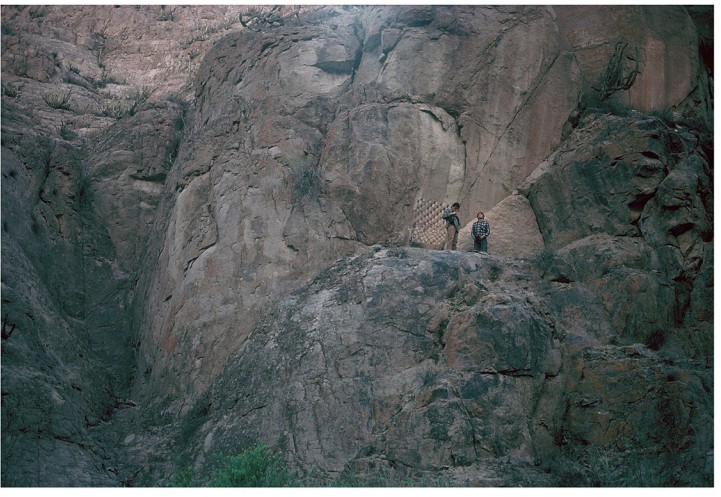
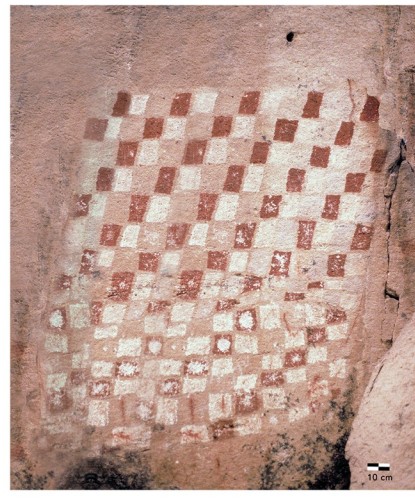

a

b

**Fig 3. Rocky cave in the Molle Grande sector, Codpa Valley, upstream from the Chaca or Vitor canyon.** (a) general view of the pictograph with red and white squares on the lower half of the steep valley wall; (b) closeup of the pictograph. (Photograph courtesy of Arthur Aufderheide†).

which is a piece of sharpened wood (11.7 cm long x 0.6 cm diameter) fastened to the distal end of the shaft with a thin fiber binding. The bow (51 cm long) was made from a single piece of cylindrical, slightly-curved wood. At the ends of the bow are remains of a cotton string; both bow and string are pigmented with red ochre.

## Standard analytical parameters and their descriptive attributes for *unku* classification

To define the standard analytical parameters and attributes associated with the production, use, repair, and discard of *unku*, we used a methodology developed by Splitstoser [62], which provides a practical framework for data collection based on a quantitative and qualitative analysis of textile attributes. Splitstoser's method recognizes the various stages of the step-by-step *chaîne opératoire* of textile production [62]. The process spans raw material acquisition and preparation, yarn spinning and dyeing, loom choice and preparation, color planning, warping and weaving, off-loom finishing, use, reuse, repair, and disposal [50]. This analytical process makes it possible to identify the different stages of the textile's life history [101].

In Table 1 we list these nine analytical parameters and 26 corresponding attributes, arranged in numerical and alphabetical order. Table 2 presents the parameters and attributes

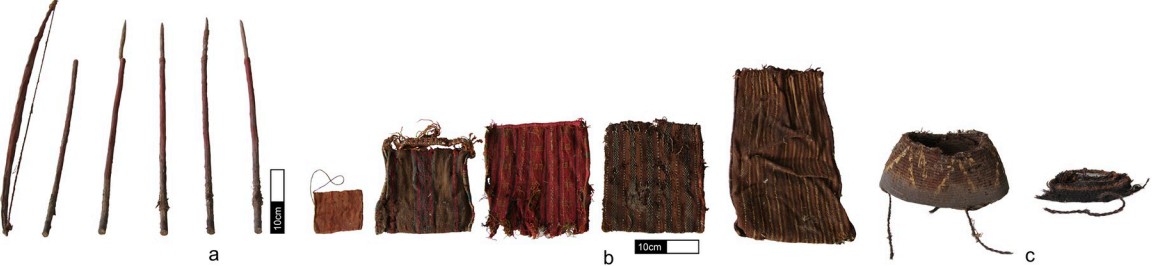

a

b

c

**Fig 4. Main objects associated with the excavated Tomb No. 1, Caleta Vitor Bay 2.** (a) bow and arrows; (b) *chuspas* and *talega*; (c) hats (photograph by Jacqueline Correa-Lau).

**Table 1. Standard analytical parameters and attributes.**

| Parameters (P) | Attributes |
|---|---|
| P1. Material Selection | 1a. Fiber type |
| P2. Spinning | 2a. Yarn type |
| | 2b. Yarn length |
| | 2c. Yarn thickness or count |
| | 2d. Yarn structure |
| | 2e. Degree of twist |
| P3. Color Selection | 3a. Color name |
| | 3b. Color Source |
| P4. Loom Selection | 4a. Horizontal loom |
| | 4b. Vertical loom |
| P5. Preparation and Set Up | 5a. Geometric shape |
| | 5b. Size |
| | 5c. Symmetry |
| | 5d. Spatial layout |
| P6. Technological-Structural Construction | 6a. Weave structure |
| | 6b. Weave density |
| P7. Design and Decorative Effects | 7a. Diamonds or rhomboids |
| | 7b. Stripes or bands |
| | 7c. Checkerboard |
| | 7d. *Tocapu* |
| | 7e. Zigzag |
| | 7f. Textures or effects |
| P8. Finishes | 8a. Types of finish |
| P9. Usages of the *unku* | 9a. Initial use |
| | 9b. Reuse |
| | 9c. Disposal |

used to study the CV *unku*. These terms are further defined in S2 Table. These features are used to identify the technical and stylistic attributes that deviated from imperial weaving canons and which are likely associated with the geographical origin of the textile.

## Results

### Case study of the Caleta Vitor Bay Inka *unku*

A detailed description of our findings for each structural element (Fig 5) is presented in S1 File and summarized in Table 2. These data allowed us to methodologically deconstruct the textile *chaîne opératoire* step by step, addressing both high-visibility and low-visibility attributes, and enabling us to distinguish morphological, technical, and stylistic state hallmarks from local ones in the CV *unku* (as noted in the last column of Table 2). In sum, the proposed methodological tool performed satisfactorily in our analysis of the CV *unku*.

## Discussion

Results obtained from our methodological tool provided us with a better understanding of the visual and technological features of the CV *unku*. We were able to characterize a series of visible traits strongly associated with the Inka State, suggesting this tunic would have been recognized as a state symbol of power and prestige at the time it was worn. If our analysis had

**Table 2. Analytical tool.**

| Standard Analytical Parameters and Attributes | | Application of the Analytical Tool to the CV *Unku* | | |
|---|---|---|---|---|
| Parameter | Attribute | Results | | |
| | | Specific detail/ characteristic | Location on textile from weaver's viewpoint | Imperial or local feature |
| Material Selection | Fiber type | Cotton | Warp 1; heading cord | Top/bottom and central support | Imperial |
| | | | Warp 2 | Complete piece | Imperial |
| | | Alpaca | Weft 1 | Selvages | Imperial or local |
| | | | Wefts 2, 3, 4 | Flange (checkered) | Imperial or local |
| | | | Wefts 5, 6 | *Pampa* | Imperial or local |
| | | | Wefts 7, 8, 9, 10 | Center or *taypi* (striped) | Imperial or local |
| | | | Repair 1 | *Pampa* (side B) | Local |
| | | | Repair 2 | | |
| 2. Spinning | a. Yarn type | Monochrome/regular | Warps 1, 2; Wefts 1, 2, 3, 4, 5, 6, 7, 8, 9, 10 and heading cords | Complete piece | Imperial or local |
| | | Monochrome/regular | Repair 1 | *Pampa* (side B) | Local |
| | | Melange | Repair 2 | | |
| | b. Yarn length | Cotton | $N_1$: 2600 cm of cotton yarn in each quadrangle | i.e., warp in each quadrangle | Imperial |
| | | Alpaca | $N_2$: 4000 cm of alpaca yarn in each quadrangle | i.e., weft in each quadrangle | Imperial |
| | c. Yarn thickness or count | Warp 1 | (c) regular (10–18); (d) 2nd order Z(2S); (e) strong (30–45˚) | Top/bottom and central support | Imperial or local |
| | | Warp 2 | (c) regular (10–18); (d) 2nd order S(3Z); (e) strong (30–45˚) | Complete piece | Local |
| | | Weft 1 | (c) very fine (30–35); (d) 2nd order S(2Z); (e) strong (30–45˚) | Selvages | Imperial or local |
| | | Wefts 2, 3 | (c) very fine (30–35); (d) 2nd order S(2Z); (e) strong (30–45˚) | Flange (checkered) | Imperial or local |
| | | | (c) very fine (30–35); (d) 2nd order S(2Z); (e) strong (30–45˚) | | |
| | d. Yarn structure | Weft 4 | (c) very fine (30–35); (d) 2nd order S(2Z); (e) strong (30–45˚) | Flange. Half-check | Imperial or local |
| | e. Degree of twist | Weft 5 | (c) very fine (30–35); (d) 2nd order S(2Z); (e) strong (30–45˚) | *Pampa* | Imperial or local |
| | | Weft 6 | (c) very fine (30–35); (d) 2nd order S(2Z); (e) strong (30–45˚) | | |
| | | Wefts 7, 8, 9, 10 | (c) very fine (30–35); (d) 2nd order S(2Z); (e) strong (30–45˚) | Center or *taypi* (striped) | Imperial or local |
| | | Repairs 1, 2: | (c) fine (30–18); (d) 2nd order S(2Z); (e) strong (30–45˚) | *Pampa* (side B) | Local |

(*Continued*)

**Table 2.** (Continued)

| Standard Analytical Parameters and Attributes | | Application of the Analytical Tool to the CV *Unku* | | | |
|---|---|---|---|---|---|
| **Parameter** | **Attribute** | **Results** | | | |
| | | **Specific detail/ characteristic** | | **Location on textile from weaver's viewpoint** | **Imperial or local feature** |
| 3. Color Selection | a. Color name | Warp 1 | (a) beige 2.5Y 7/4; (b) natural | Top/bottom and central support | Imperial |
| | | Warp 2 | (a) beige 2.5Y 7/4; (b) natural | Complete piece | Imperial |
| | | Weft 1 | (a) dark brown 2.5Y 3/3; (b) dyed | Selvages | Imperial or local |
| | | Weft 2 | (a) white 2.5Y 8/1; (b) natural | Flange (checkered) | Local |
| | | Weft 3 | (a) very dark brown 2.5Y 2/1; (b) dyed | | |
| | | Weft 4 | (a) dark brown 2.5Y 3/3; (b) dyed | Side B. 2nd and 4th row, 4th quadrangle from bottom to top | Local |
| | | Weft 5 | (a) light garnet red 10R 3/6; (b) dyed | *Pampa* (side A), lower section | Imperial |
| | b. Color source | Weft 6 | (a) dark garnet red 10R 2/3; (b) dyed | *Pampa* | Imperial |
| | | Weft 7 | (a) white 2.5Y 8/1; (b) natural | Center or *taypi* (striped) | Local |
| | | Weft 8 | (a) very dark brown 2.5Y 2/1; (b) dyed | | |
| | | Weft 9 | (a) medium brown 2.5Y 5/6; (b) dyed | | |
| | | Weft 10 | (a) very dark brown 2.5Y 2/1; (b) dyed | | |
| | | Repair 1 | (a) dark brown 2.5Y 3/3; (b) dyed | *Pampa* (side B), upper section | Local |
| | | Repair 2 | (a) very dark brown with white fibers 2.5Y 2/1; (b) dyed | *Pampa* (side B), upper section | Local |
| 4. Loom Selection | a. Horizontal loom | Not applicable | | Not applicable | Not applicable |
| | b. Vertical loom | Yes | | Not applicable | Imperial |
| 5. Preparation and Set Up | a. Geometric shape | Rectangular along the width (width greater than height) | | Not applicable | Imperial |
| | b. Sizes | 80 cm high x 182.5 cm wide | | Piece fully stretched out | Imperial |
| | | 80 cm high x 35 cm wide (side B) | | Checkered section (complete section) | Imperial |
| | | 26 cm | | Fully open neck slot | Imperial |
| | | Between 7.4–9.5 cm high x 8.2–10 cm wide | Pattern repeat in two alternating colors | Flange (checkered) | Imperial |
| | | Between 4.0 and 7.0 mm wide | Pattern repeat in four alternating colors | Center or *taypi* (striped) | Local |
| | c. Symmetry | Symmetric in shape and asymmetric in color | | Vertical and horizontal | Local |
| | d. Spatial layout | Heading cord | | Top/bottom | Imperial or local |
| | | Checkered | | Flanges | Imperial |
| | | Plain area | | *Pampas* | Imperial |
| | | Striped diamond | | Center or *taypi* | Local |

(*Continued*)

**Table 2.** (Continued)

| Standard Analytical Parameters and Attributes | | Application of the Analytical Tool to the CV *Unku* | | |
|---|---|---|---|---|
| Parameter | Attribute | Results | | |
| | | Specific detail/ characteristic | Location on textile from weaver's viewpoint | Imperial or local feature |
| 6. Technological-Structural Construction | a. Weave structure | Heading Cords | Top/bottom and central support | Imperial or local |
| | | Interlocking tapestry | Whole piece | Imperial |
| | | Discontinuous warps | Neck slot | Imperial |
| | | Slit tapestry | Intersection between *pampa* and striped diamond | Imperial |
| | | Wefts grouped in 3/3 | Joining of two parts (pieces) of the textile | local |
| | b. Weave density | Warp 1 | 2 threads at 3 mm | Top/bottom and central support | Imperial |
| | | Warp 2 | 26 threads per cm | Complete piece | Imperial |
| | | Weft 1 | 8 passes at 3 mm | Selvages | Imperial or local |
| | | Weft 2 | 40 passes per cm | Flange (checkered) | Imperial |
| | | Weft 3, 4 | 48 passes per cm | | |
| | | Weft 5 | 44 passes per cm | *Pampa* | Imperial |
| | | Weft 6 | 68 passes per cm | | |
| | | Weft 7 | 32 passes at 7 mm | Center or *taypi* (striped) | Local |
| | | Weft 8, 9, 10: | 28 passes at 7 mm | | |
| | | Repair 1 | 14 passes per cm | *Pampa* (side B) | Local |
| | | Repair 2 | 15 passes per cm | | |
| 7. Design and Decorative Effects | a. Rhomboids or diamonds | Yes | Half rhomboid or "V" shape which forms a breastplate. Side corners (shoulder height) are truncated. | A complete rhomboid is formed when the piece is fully extended. | Imperial |
| | b. Stripes or bands | Yes | Regular stripes based on a 4-color pattern repeat (one of the colors is repeated) | Center or *taypi* (rhomboid) | Local |
| | c. Checkered | Yes | 4 rows of 10 quadrangles and two alternating colors | Flange | Imperial |
| | d. *Tocapu* | Not applicable | | Not applicable | Not identified[a] |
| | e. Zigzag | Not applicable | | Not applicable | Not identified[a] |
| | f. Textures and/or effects | Eccentric weft | Visual curvilinear and ribbed effect | *Pampa* (side A), lower section | Imperial or Local |
| | | Lazy lines | Diagonal lines | *Pampa* (side A), upper section | Imperial or Local |
| 8. Finishes | a. Type of finish | Overstitch | | Neck slot (side A) | Imperial or Local |
| | | Figure-eight stitch | | Not identified | Not identified[a] |
| | | Zigzag-reinforced stitch | | Not identified | Not identified[a] |
| | | Fishbone stitch | | Not identified | Not identified[a] |
| | | Heading cords | | Top/bottom | Imperial or Local |
| 9. Usages of the *Unku* | a. Initial use | Original | | Not applicable | Local |
| | b. Reuse | Repairs | Two in the *pampa*, very fine weave which imitates the structural weave parameter. One of these is more visible due to its *mélange* yarn | Pampa (side B), upper section | Local |
| | c. Disposal | burial | | Not applicable | Local |

[a]Imperial feature not identified in the CV *unku*.

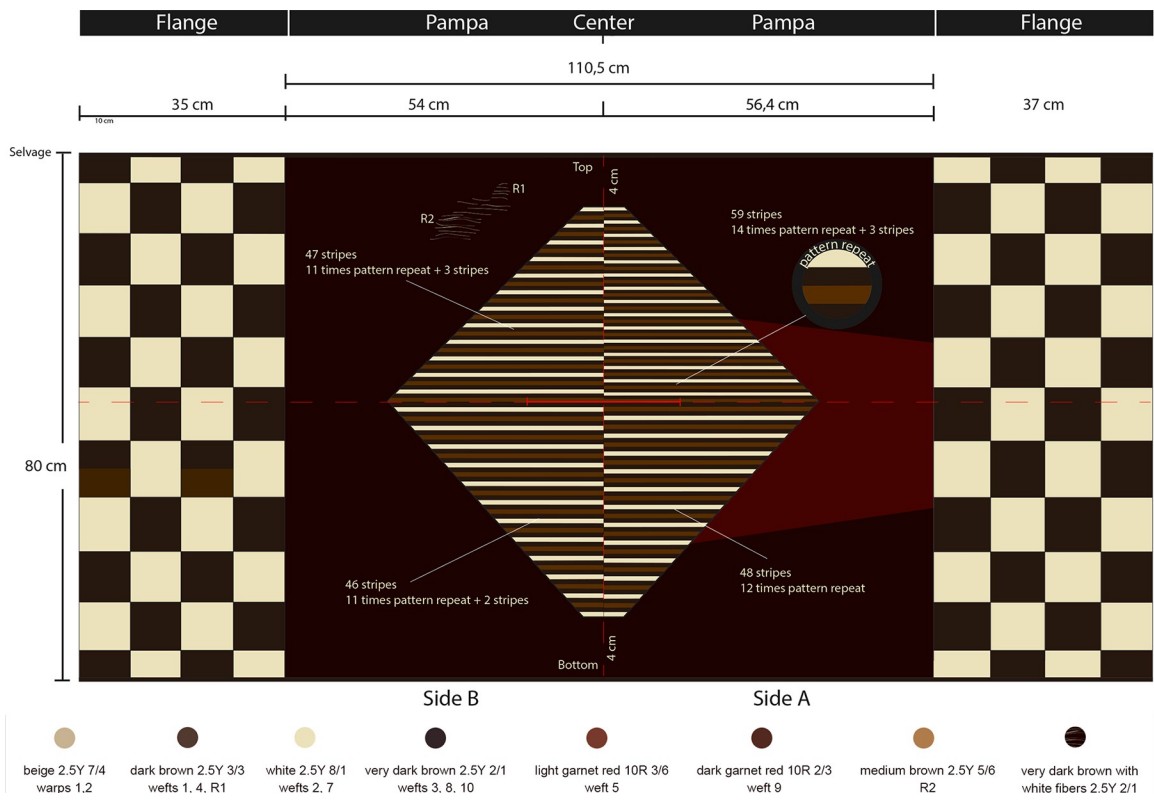

**Fig 5. Description and detail of the CV *unku* (illustration by Pamela Aravena).**

focused solely on identifying and defining the visual attributes of Inka traits (*techne*), we would not have identified a series of traits unique to this piece and deriving, we propose, from local traditions (*metis*). The dialectical processes we identified between *metis* and *techne* must have been a source of tension between the skilled weavers and the technical and stylistic demands of the state. Our analysis provides a deeper understanding of the processes involved in textile production under the pressures of adhering to state canons. We also shed light on the identificatory character ascribed to textiles by their makers that helps reveal their geographic origins.

The CV *unku* is especially remarkable for its stylistic singularity, reflecting the technological-structural syncretism that gave rise to this particular provincial Inka *unku*. Brugnoli et al. [102] point out that Andean garments embody a narrative and, according to Wobst's theory of style [103], would have been designed in such a way as to be seen from afar to convey syncretic symbolism. In this same vein, other iconic material goods from the Inka period, such as ceramic *aryballos*, wooden *queros* [104], *chuspas* [27], tunics [12], *ushnus*, and monumental architecture, display the hallmarks or identity traits of local cultures, distinguishable from those of the empire.

In one sense, the CV *unku* represents well-known and highly-visible Inka imperial standards (*techne*). This includes a long rectangular morphology and lack of sleeves. The measurements are within the established averages for width [between 72–79 cm; 6] and length [between 79–110 cm; 44]; see also Mary Frame [12]. The *unku's* interlocking tapestry technique is certainly of highland origin, as are its checkerboard design and "V"-shaped yoke, which forms a diamond when the piece is laid out fully. The diamond design may somehow reflect the Inka cosmovision of the world divided into four zones or *suyus*. On the other hand,

the slit tapestry technique at the intersection of the *pampa* and diamond seems to refer to a coastal origin [12].

The CV *unku* presents several low-visibility structural and stylistic variants that we attribute to local/regional practices (*metis*), possibly of coastal origin. First, the textile was woven with cotton warp and camelid fiber wefts. Because imperial *unku* could have either camelid fiber or cotton warps, this latter feature is not diagnostic of an *unku* with a coastal origin; however, there is a strong association between cotton and coastal textiles. Although the CV *unku* was conceived as a single piece, it was made in two parts. This becomes visible where the sections converge, denoted by the heading cords that add structure to the shoulder area of the textile. At the shoulder seam, there is a small incongruity in the alternating colors of the diamond-enclosed stripes and pattern repetition, which appears to be the result of more than one weaver working together, as Rojas and Hoces de la Guardia [85] have proposed for other Inka textiles. Apart from the change of hands that may have led to these kinds of mismatches, they might also have resulted from a lack of rigor in the finishing process. Alternatively, it could be suggested that this "error" came about due to an intentional decision on the part of the weavers to impart local/regional practices, or *metis*, to deviate from state norms. That being said, the evidence would not have been immediately visible, neither to the wearer nor the observer, given its position at shoulder height.

The solid stripes that make up the diamond are organized based on a repetition of natural, contrasting colors within the brown range. This stripe pattern is regularly recorded in other woven textiles such as bags (*talegas*, *costales*, *chuspas*, and *inkuñas*) and tunics from earlier periods excavated along the southern coast of Peru and throughout northern Chile. The use of this color scheme and its adaptation to the CV *unku* is a highly visible aesthetic feature. For this reason, we see this decision as reflecting a choice made by local coastal weavers, who, though adhering to general state patterns, still managed to leave their own hallmark, thereby characterizing and marking the piece as a provincial Inka *unku*.

Similarly, the colors of the checkerboard squares present variations in range that go beyond those of the state. They are not the standard black-and-white squares as described by John Rowe [7]. Indeed, we identified the black color, macroscopically and microscopically, as dark brown to very dark brown. At first glance and from a distance, the very dark brown color could easily have passed for black from the point of view of an observer engaged in certain activities when the piece was worn. On B side (from the wearer's point of view), two squares were woven half in very dark brown and half in dark brown. Rather than technical errors or design mismatches, these digressions are, in our view, the product of deliberate actions by local weavers. This characteristic of two squares with chromatic duality in the brown range is exemplified on a large scale (i.e., the entire surface of the textile) in two *unku* defined as provincial in the collection of the Ethnological Museum of Berlin (VA 16630 and VA 4576) [105]. Another case, but with curved figures, is observed with a duality of blue range (VA 62696). The CV *unku*, a piece that integrates imperial and provincial morphologies, technologies, and styles, is a perfect example of syncretism, which transforms it into a local product. The detail of the chromatic duality in a single square of a checkered surface has not been noted in any other *unku* of the BW type, with which this piece would be associated. Therefore, we suggest investigating this feature as an element of the creation of local weavers who, following the state templates, introduced distinctive hallmarks. As for the prominence of this feature, although it is located on the front flange of the garment, due to slight tonal differences, it is not readable. Similarly, it occurs with the intersection of the stripes at shoulder height, which are in less conspicuous positions. We identified both features as having low-visibility.

Another technical decision with subtle visual effects was the change in shades of red produced by the "lazy lines" [88] which occur diagonally across the red area. Also, the tightly

packed wefts in the structural area produced a ribbed visual effect that enhances contrast in the warps. The latter effect may be due to the use of S(3z) warps, which are thicker and less dense (10–18 per cm) than the weft yarns. Furthermore, some of the warps appear to be "floating" above the flat red area. In the first instance, we presumed these were technical errors. However, given these traits are repeated on both sides of the piece, we consider this to have been an intentional *a priori* effect introduced by its makers, which we interpret as another attribute of the CV *unku* that was the result of local weaving practices.

Repairs to the piece were made by intricately and delicately emulating the original technique. As such, they are not immediately apparent and would have been barely perceptible to observers. For this careful repair work, the mender used composite yarns of different colors to those of the original textile, thus disclosing the intention to modify the conservation of the piece and unveil his hallmark as a mender. The repairs were intended to prolong the lifespan of the piece and thus give way to the second usage (reuse) of its life history. The CV *unku* was a singular piece in which, at first glance, state and provincial canons are identified, which thus made it an object valued locally and representative of a social, cultural, geographical, and temporal context. In this sense, this piece was transformed into an inheritable and desirable object given its ideological and political charge, which was to be maintained in perpetuity through reparations, thus endorsing the position of the character or family lineage in which the piece fell, to the extent that it continued to be part of the structure of the political relationship between the state and this local community.

Given the ecological constraints in developing their economic programs, the hyper-arid coast of northern Chile was somewhat indirectly linked as a sociopolitical unit of the Inka State compared to those of its inner core. Its most important resources were the abundant seabird guano reserves along the northern coast of Chile, where guano deposits are still visible at Caleta Vitor Bay, and the rich silver mine of Huantajalla, south of Caleta Vitor Bay. Control and exploitation of these key resources generated traffic routes along the coast and toward the hinterland [98, 100, 106]. Caleta Vitor Bay was therefore one of the nodes within this territorial economic sphere and was likely part of the guano production and distribution network, which was coordinated by the Inka State and essential to agricultural activities within the desert and beyond. According to Williams et al. [67], failure to reach negotiated agreements with the state meant that local communities remained outside imperial networks, which was clearly not the case with the community at Caleta Vitor Bay.

The CV *unku* and accompanying objects recorded from Caleta Vitor Bay Tomb N˚ 1 may reflect broader negotiations and agreements that took place between Inka State officials and local Caleta Vitor Bay authorities to access and control guano extraction and the movement of silver mined from Huantajalla. This relationship would also have brought with it a series of obligations to the state as well as provisions for local communities. In these negotiations, there was no commitment on the part of the Inka to invest in any monumental architecture projects such as the *ushnu* erected in several other localities scattered along the interior coastal valleys. In fact, the material culture that defined the relationship between the state and the Caleta Vitor Bay social group would have been non-monumental in nature and operated primarily through symbolic elements. Within this ensemble, the CV *unku* stands out for its eloquence.

If it is accepted that the CV *unku* was the result of local production, we might suggest that the garment was produced under the state *mit'a* system, which according to Murra [14] was one of the main work obligations demanded of local communities by the state. This means that the products of this local *mit'a* were state property, which were distributed inside or outside the territory, depending on the state's needs.

The CV *unku*'s checkerboard design has been interpreted as a military symbol. The idea that the CV *unku* may have been worn by a local member of the state military is also supported

by some of the associated archaeological materials (i.e., breastplate and possibly the bow and arrows). Tentatively, we estimate that this function (military service) may have corresponded to another *mit'a* obligation incurred by the Caleta Vitor Bay community. In Cusco, according to Murra [14], military garments were produced by women and given by the Inka government to men of military age, an endowment that represented a state reward. These tunics may have been produced by relatives of the soldiers. Similarly, we estimate that the state may have gifted the locally produced CV *unku* to this military figure as a prestige object befitting his position. In this process of giving and receiving, the state enacted the redistribution system of *mit'a* produced objects with the community of Caleta Vitor Bay. The action of giving helped bring negotiations and agreements to a successful conclusion, paving the way for integration and fostering collaboration between the community and the state.

## Conclusions

Regarding the origins of the CV *unku*, given its recognizable morphological, technological, and stylistic features, we are inclined to suggest it was made by weavers who were very well versed in the textile traditions of the southern coast of Peru and northern Chile, which constituted its identificatory attributes. These textiles of local origin, however, had to be stylistically and functionally equivalent to those produced by state weavers in Cusco and other state production centers. Consequently, local weavers from an unknown locality in the Arica or Tarapacá provinces would have endorsed the state's conventional traits to ensure the CV *unku* functioned stylistically and structurally as an item of military clothing. However, the local weavers were able to add their own designs (e.g., stripes) while applying enough Inka details to allow the wearer to project his role as a state dignitary. Thus, the CV *unku* embodied the social, economic, and political relations and obligations negotiated between the state and local communities.

The textile pieces provided by the state formed part of the collective memory of social groups and served, intrinsically and extrinsically, to maintain the reciprocal tradition of giving and receiving coopted by the State. However, they also demonstrate the flexibility that local communities had in being able to introduce their own set of idiosyncratic variations, modifications, and substitutions. At the same time, local weavers understood how critical it was to uphold certain immutable and essential fixed traits of Imperial importance, ensuring that their technical actions would not compromise the pieces' symbolic function. This syncretic condition of the textile could also be repeated in other goods produced by and for the state. For example, despite the rigid production processes of ceramics, artisans managed to introduce technical and aesthetic variations that resulted in a diversity of vessels defined as local or provincial Inka.

In conclusion, the signals, variations, and visual and haptic effects of the CV *unku* demonstrate the existence of local agents operating within an asymmetric and non-unidirectional system of state-community relations. The tension embodied by weavers in having to explicitly reproduce state patterns, and mask or subtly unveil their local hallmarks are well represented and expressed in this *unku*. This tension has also been replicated in other woven pieces (e.g., *chuspas*). Each one of these textiles preserves a unique and inalienable life history, representing the vestiges that attest today to the dialectic processes at work in the development of the Inka State.

## Supporting information

**S1 Table. List of *unku* from around the world.** Register of *unku* located in museums in Europe, the United States of America and South America.
(DOCX)

**S2 Table. Standard analytical parameters and attributes.** Description of the 9 analytical parameters and 26 attributes.
(DOCX)

**S1 File. CV *unku* parameters and attributes.** Supplementary description that characterize the CV *unku*.
(DOCX)

**S1 Fig.**
(TIF)

## Acknowledgments

To Persis Clarkson for her careful revision of the English; to Benjamin Ballester for informing us about two *unku* excavated and photographed by Max Uhle in 1916, held at the Ibero-Amerikanisches Institut and Ethnological Museum of Berlin; to Lena Bjerregaard for proving further information about the collection of this *unku*, and Mauro Bologna for refining a formula to calculate yarn length. To Natalia Aravena and Arnoldo Vizcarra for the optical microscopy and electron microscopy images of the *unku* yarns at the Laboratorio de Bioarqueología, Instituto de Alta Investigación, Universidad de Tarapacá, courtesy of Bernardo Arriaza. To the anonymous reviewers and editors for their fruitful comments and suggestions that substantially enriched this work.

## Author Contributions

**Conceptualization:** Jacqueline Correa-Lau, Ester Echenique, Calogero M. Santoro.

**Data curation:** Jacqueline Correa-Lau.

**Formal analysis:** Jacqueline Correa-Lau.

**Funding acquisition:** Ester Echenique, Calogero M. Santoro.

**Investigation:** Jacqueline Correa-Lau, Carolina Agüero, Tracy Martens, Calogero M. Santoro.

**Methodology:** Jacqueline Correa-Lau, Jeffrey Splitstoser.

**Project administration:** Jacqueline Correa-Lau, Calogero M. Santoro.

**Supervision:** Jacqueline Correa-Lau, Carolina Agüero, Tracy Martens, Calogero M. Santoro.

**Validation:** Jacqueline Correa-Lau, Carolina Agüero, Jeffrey Splitstoser, Ester Echenique, Tracy Martens, Calogero M. Santoro.

**Visualization:** Jacqueline Correa-Lau, Carolina Agüero, Calogero M. Santoro.

**Writing – original draft:** Jacqueline Correa-Lau, Carolina Agüero, Jeffrey Splitstoser, Ester Echenique, Calogero M. Santoro.

**Writing – review & editing:** Jacqueline Correa-Lau, Carolina Agüero, Jeffrey Splitstoser, Ester Echenique, Tracy Martens, Calogero M. Santoro.

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
