## [Decision Letter · Decision Letter 0]

17 Aug 2022

PONE-D-22-00867Inka Unku: Imperial or Provincial? State-Local RelationsPLOS ONE

Dear Dr. Correa-Lau,

Thank you for submitting your manuscript to PLOS ONE. After careful consideration, we feel that it has merit but does not fully meet PLOS ONE’s publication criteria as it currently stands. Therefore, we invite you to submit a revised version of the manuscript that addresses the points raised during the review process. Each of the three reviewers indicate that major revisions are required before your work will be suitable for publication. Reviewers 1 and 3 provide detailed comments on areas of the manuscript that need to be reworked for clarity and consistency. Please take all of the reviewers' comments, suggestions, and criticisms into account while making revisions.

We look forward to receiving your revised manuscript.

Kind regards,

John P. Hart, Ph.D.

Academic Editor

PLOS ONE

Journal Requirements:

Reviewers' comments:

Reviewer's Responses to Questions

**Comments to the Author**

1. Is the manuscript technically sound, and do the data support the conclusions?

Reviewer #1: Yes

Reviewer #2: Partly

Reviewer #3: Partly

2. Has the statistical analysis been performed appropriately and rigorously? 

Reviewer #1: N/A

Reviewer #2: I Don't Know

Reviewer #3: N/A

3. Have the authors made all data underlying the findings in their manuscript fully available?

Reviewer #1: Yes

Reviewer #2: Yes

Reviewer #3: Yes

4. Is the manuscript presented in an intelligible fashion and written in standard English?

Reviewer #1: Yes

Reviewer #2: No

Reviewer #3: Yes

5. Review Comments to the Author

Reviewer #1: 1. Is the manuscript technically sound, and do the data support the conclusions? Yes, the information provided on the CV unku (file S2) is sound data that delineates relevant unku/tunic features. The context of the unku find (at a CV burial) and references to technique + design support argument that local elements become factors in addition to state (unku) standards.

2. Has the statistical analysis been performed appropriately and rigorously?

N/A for statistical analysis. Yes, the review of the one unku sample is rigorous and helpful to considering more about deviations from Inka standards or how “syncretic” aspects can come forward but as the authors note what is offered is a semi-quantitative and qualitative analysis of one unku more than a statistical analysis.

3. Have the authors made all data underlying the findings in their manuscript fully available? Yes, the separate data sheet on the CV unku is helpful in addition to the tables

4. Is the manuscript presented in an intelligible fashion and written in standard English?

Yes, although there is significant room to review content for clarity here and there. See comments below in #5.

5. Review Comments to the Author

Regarding the paper’s intent to “help us visualize the relationship between technological action and propagation of state power/ideology,” good observations are made regarding how local social agency (via technological aspects) is expressed. However, there are several areas where sentences or paragraph organization can be clearer. With regard to the parameters, for example, of “visible and low-visibility” attributes and the terms “technological” and “aesthetic,” sometimes the reader loses their way. The distinction of what is a technical attribute and what is an aesthetic attribute, for example, or how technology and style interface is not always precise and sections could be reviewed for more precision in word usage. Likewise, with content related to whether (local) adjustments to state standards are visible or not, there is a section (lines 471-498) where the reader loses sight of what the authors aim to do in terms of (as stated in line 87) bridging “the knowledge gap between visible and low-visibilty attributes.” I note in comments at end a text by Arnold and Espejo that is a good reference for how technology, structure, and motifs/aesthetics can be discussed in correspondence with one another. I don’t see it referenced in the bibliography and it would make sense to include it.

The tables provide relevant information re: standardization aspects in Inka unkus and the particular traits of the subject unku but there are a couple of things that might make the table 2 more digestible. One thing, for example, is that there could be color coding or something introduced in the Table 2 to highlight where CV unku’s features deviate from the Inka standard. This would be a helpful visual assist to the reader.

Throughout there are parts that bring up questions re: word choice or whether content is coming across clearly enough—I've made some notes below

Section 1. Introduction

From the Intro’s parag. 1 where authors note an attention to “technical-formal-aesthetic” elements but then later at Line 89 the reference is to “technological-structural-aesthetic” attributes. Should be consistent, formal is not equivalent to structural …or the equivalence authors are making should be clarified

Lines 93-96—can be clearer—reference is made to “the unku” but it also seems reference is being made to unkus in general here, (e.g. “in some cases”) so which is the subject (one or all unkus available for study) of these sentences could be clearer.

Section 2. Inka unku, imperial material imagery

Lines 101-102 – re: “formal-functional, structural-aesthetic”-- the hyphenated relationships are a little confusing. Can it be more straightforward? E.g. technical (structural), aesthetic, functional, and socially recognized elements

Line 105—last sentence seems out of place, perhaps belongs further up somewhere

Lines 107-116 seem out of place here or unnecessary

Line 120—re: not sure this is best word choice re: “disused”

Line 122—sentence “During the early colonial period…” seems irrelevant?

Line 125—re: wording/word choice “’man-and-clothing’ relationship” is a little awkward

Lines 131-134: these first sentences don’t really correspond to the paragraph, could be removed

Line 136: suggest simplifying “consisted of six styles. This typology has been used to broaden…”

Line 146: could re-phrase/check wording “developed politically”

Line 153: can be clearer re: mention of one-way perspective here, since previous sentence refers to different perspectives. Can this sentence be re-phrased for clarity

Line 155: Sentence re: “We believe that…” is very crammed with wording (of too many associations) and tries to do too much. Suggest breaking into separate sentences or re-phrasing for clarity. In other words, following the beginning of the next sentence, how are all these “points of convergence”?

Line 159: Sentence “In other words” seems unnecessary here and disrupts rather than helps

Lines 164-168: sentence “These creative processes…” can be revised for clarity—what is the takeaway here

Line 173: re: word choice-- “patterns” may not be the best word choice? State criteria? See note in 'Materials and Methods'

Line 173: A couple of things are unclear here. Does the word “unku” here refer to the specific unku in this study or to unkus of this coastal region or to standard Inka unkus in general? Can clarify… the unku that is the subject of this study shares the camelid warp and weft criteria seen in Inka checkerboard unkus…but not all standardized state unkus (as referenced by Anne Rowe and John Rowe) have camelid weft and warp…some examples (esp. diamond waist) have cotton warp

Line 179—if the previous sentence mentions the concealed gesture of multiple wefts, why does this sentence begin with “similarly” but then refer to the “simple weft” type that is recognized along Arica coast? It’s a little confusing.

Lines 203-205—re: “…the search for… Inka state conventions in garments has been limited to analysis of their visible side”…unclear sentences in these 2 lines…sentence content can be clarified as implication is that no significant analysis has been made beyond just the visible

Line 212—re: word choice “sceneries”

Section 3. State and provincial unku. Form, function, structure, and aesthetics

Line 237. Can clarify—ref. Rowe, in some standardized state unku examples, we do see cotton warp

Line 256 – re: “special looms, hung on the wall,”….rather the suggestion from Rowe, Cobo, etc. is that the vertical looms were propped against a wall/ set upright against a wall…

Line 281—can be clarified, when you talk about the ahuaqui/awaki, the suggestion is that this is a reference to rhomboid/diamond motifs in general (according to the paragraph context) but the term is often used to refer to the area around the neck (the yoke) that may have stepped edges outlining a rhomboid… Or where is the citation for the ref. to Wari ahuaqui?

Line 289—unclear…what is meant by “in the case of the plain stripes…located around the arm slits”? What are the “plain stripes”? Is this a reference to woven stripes, as seen in the diamond cowl of the subject tunic, in which case the word would not be “applied to pieces” but “woven into the garment”…or is it reference to the embroidered striped cross-knit loop stitching (which would be “applied”) seen often on Inka tunic armholes and bottom edges?

Lines 319 and 322—unclear…but zigzag embroidery does not seem to be an element that lends the tunics more structure, which might be said of the cross-knit loop stitching at the hems or the figure-eight stitches on the side seams. Also, the suggestion in this paragraph is that all standardized unkus have zigzag stitching, but this is not the case…

4. Materials and Methods

Regarding Sub-section “Standard Analytical Patterns / attributes/classification”- the first several paragraphs have a lot of repetition and the content could be streamlined. Suggest editing these paragraphs down to clarify main points. Everything could be summed up more or less in a few sentences instead of a few paragraphs: “Our method is based on semi-quantitative and qualitative analysis: we define a set of standard analytical patterns and their attributes on the Inka unku which, in turn, are understood as technical and aesthetic “hallmarks” of production. Framing our analysis within modes of production, we follow Splitstoser’s methodology, which recognizes…Our attempt to deconstruct process in the operation chain helps us understand….textile production norms.”

Question: why the word “patterns” to describe the analytical criteria? Not sure it’s the best word choice. “Patterns” is too resonant with how we would consider surface aspects of a textile (design/aesthetic, display/arrangement). Can another word be used to describe the considerations? “Criteria”? “Aspects”? “Elements”? “Parameters”?

-Re: lines 432-433…what is the difference between “utilitarian features” and “functional particularities”?

-Re: line 432—“they were developed specifically for analyzing the unku..”—most of the “patterns” seem to refer to unku as a type rather than this particular unku…can this be clarified in this section?

This can also be clarified in the language used in Table 1, each section of which should be reviewed for language clarity and precision re: study parameter—e.g. sentence “we have divided the piece into separate parts,” can be clearer b/c don’t the authors mean “we have divided the Inka unku (as a type) into separate parts”? In other words, this is not about this specific piece but a general assessment of the type…but it’s not clear in the language.

Further issues regarding language clarity—for example, in “Attribute 7a. Diamonds or rhomboids,” the use of the word “structure” doesn’t seem appropriate here. The geometric shapes form the composition by outlining spatial areas. When the word “structure” is used, it confuses terms, since “structure” really connotes the interior/technical elements not the surface, design elements.

5. Results

Table 2

Re: “Location of textile from weaver’s viewpoint” revise “Location on textile…”;

Re: 2. Spinning, a. Yarn type, clarify: “Simple: regular/monochrome”

Re: 2. Spinning--the break up of the table here with parts c. through e. distributed differently—both on the horizontal and vertical axes-- and re: 3. Color Selection. Would it be useful to have this differently organized content put into a separate table to avoid having this irregularity in table organization?

6. Discussion

Lines 457-458—these two sentences “More specifically” and “There must have” seem to disrupt the general gist and don’t serve the paragraph

Lines 489-490—the suggestion that discrepancy in the alignment of the two parts of the large rhomboid is a “gesture of transgression” veers a bit from how the article frames local attributes in relation to state standards…should this idea of transgression be brought up sooner?

Line 471 –498 –between these lines the arc of the argument varies from suggesting that local weavers were making [local] adjustments that were highly visible or also making adjustments that were not immediately visible. Content could be more carefully argued here. Re: the point that the unku carries local tradition and skill/experience (metis), was it meant to be seen or not? The takeaway gets muddled in these paragraphs

Line 505—can clarify re: the half squares at the lateral selvages… the suggestion that half squares are a local anomaly but this does appear in many of the standardized BW unkus that John Rowe reviews

Lines 548-551—this content seems relevant as something to underscore earlier in the article as well

Line 559--word choice, not “printed on the clothing” but rather introduced into/woven into /embedded into the woven structure of the cloth…

Line 564-565—points being made could be clearer, or seem to disrupt other content flow

*Throughout double check for spelling, punctuation--, eg p. 28 table 2 heading– typo in “application”; sometimes “state” is capitalized, sometimes not…

Re: recommend including/using these additional sources:

Mary Frame, “Chuquibamba: A Highland Textile Style,” Textile Museum Journal, 1997-1998, Vols. 36, Washington DC. Frame’s essay regarding provincial Inka tunics may be of interest as she discusses various aspects of textile production from the Chuquibamba textile tradition and contextualizes them against Inka standards to a degree. Also, at parts, she considers how fabric-making technology informs design in textiles (contrast to this article’s interest in pointing to chains of operation at the local level as being influential over design aspects).

Denise Arnold and Elvira Espejo, El Textil Tridimensional: La Naturaleza del Tejido Como Objeto y Como Sujeto, Edition: Serie Informes de Investigación, II, No. 8, Fundación Xavier Albó and Instituto de Lengua y Cultura Aymara, La Paz, 2013.

See pp. 118-130 of the book--This article’s bibliography references Denise Arnold’s Textil y Tributo but I would also point to this text which discusses weaving tools and practices within the notion of chaine operatoire, particularly, that may be relevant to this article. For example, although in an ethnographic context and within an Aymara context, the authors refer to what goes into the color selection process within the operation chain fairly in-depth, referencing musa waraña. This may be useful to this article for elaborating on some points re: how color choices are made/were made perhaps in manner applicable to the question of operation chain re: color and Inka context, etc.

See also pp.183- 184 in same book. These paragraphs tease out question of how to think about style in relation to technology and may be useful, would be relevant to cite.

Also, the bibliography points to the text “An Inka Unku from Caleta Vitor Bay, Northern Chile,” written by some of this article’s same authors, with citation/note in Attribute 5b. That text’s attention to the same unku and question of Inka investment/stamp on the CV unku garment seems pertinent to this article’s inquiry and probably should be given further mention, e.g. stating how this extends/expands/or deviates from that study.

Reviewer #2: The authors present a fascinating and potentially very important argument: that Inka unku, paradigmatic symbols of imperial power, are made locally in certain cases. This would be a welcome addition to this area of research. That said, the article would be strengthened by clarifying and quantifying key aspects of the argument that at the moment are either unsupported or clouded by extraneous detail.

For example, at the outset the authors argue that the Inka "imposed technical and aesthetic canons" for the production of the tunics. How did they do this? It would be good to note the data supporting this statement. Furthermore, how does this then play into this article's argument that this tunic was outside (?) of this control? Or, if not outside of this control, how were these canons shared/imposed?

Most importantly, however, the crux of the argument lies in the presumed fact that the tunic was made locally. This is an essential piece of the argument and it needs to be presented in a clearer, more forceful way. The article will be strengthened by cutting back on jargon and presenting the data in a straightforward way. At the moment it reads more like a discursive thesis than a scientific journal piece.

I encourage the authors to tighten up their manuscript and re-submit as the research is of much merit.

Reviewer #3: On page 5, the authors write that their objective is “to bridge the knowledge gap between visible versus low-visibility attributes. To this end, we created and defined standard analytical patterns and their technological structural-aesthetic attributes to analyze the unku. This theoretical-methodological tool also serves to evaluate the possible existence, or not, of technical and/or aesthetic differences in the unku, depending on the status of the wearer.” This objective, and the relationship between it and the specific analytical approach taken to characterize the different variables of the unku, remains deeply confusing to me. First, I do not know what a “theoretical-methodological tool” is; theory refers to a conceptual framework used to interpret patterns in the archaeological record that asserts specific relationships between variables, and that can be objectively tested against predicted patterns in the record. To the extent that a theoretical framework is offered, this draft perhaps predicts variability between textiles’ attributes that can be attributed to, or used to identify, local weavers’ agency. Methodology refers to a specific set of sampling and data collection techniques that can be used to collect the data needed to objectively test specific hypotheses generated by the theoretical framework. If the analytical form in Table 2 is what the authors as referring to by the phrase “theoretical-methodological tool”, alternatively referred to on page 20 as “a theoretical practical framework for data collection”, it needs to be explained how this particular spreadsheet of attributes derives from the theoretical approach they advocate for: how, for example, the assertion that local weavers used specific technical or aesthetic differences to indicate the status of the wearer. It is also unclear why a checkerboard unka (male tunic) buried with two young women in a provincial burial is appropriate or sufficient data to test this hypothesis.

On page 5, the authors assert that “he unku were woven and distributed from Cusco across the entire territory under Inka control.” This is directly contradicted by the statement on pages 8-9 (emphasis added) that “local weavers introduced their own hallmark in the form of multiple wefts, a traditional textile technique that became a concealed technical gesture used in textiles during the Inka period [47-49]. Similarly, continual use of the simple weft has been recognized along the Arica coast.”

Indeed, I believe a better statement of the paper’s main contribution appears on page 10, where the authors write: “we have developed a decoding tool for the unku with the aim of distinguishing state from local hallmarks, thereby revealing the syncretic complexity of these iconic tunics.” If other scholars have assumed that unku were centrally produced in Cuzco and distributed throughout the empire, then your study provides additional support to Williams et al, and others’ work demonstrating local production of imperial-style ceramics, but extended to weaving production. Consider reframing the article around this narrower, but clearer objective.

I also believe that the very long list of complete attribute descriptions given in Table 1 can be selectively condensed. I would much rather, for example, have the full list – along with the form given in Table 2 – put on a website somewhere. The space could then be devoted to explaining with 2-3 specific examples how local weavers’ agency would produce different values in specific attributes than centrally-controlled or Cusqueño weavers’ textiles. In other words, the article could use a much fuller and more specific, detailed explanation of how specific attributes in the form are useful for identifying local weavers’ choices or hallmarks. Using the form, or only a select subset of attributes from the form, to show distinct differences between the Caleta Vitor unku and one or more unku that were clearly made by Cusqueño weavers, would provide a stronger argument for the form’s utility in testing hypotheses about state vs local differences in the chain of operations. The authors begin to do so on pages 34-36 in the discussion section, but I would like to see a much more detailed dive into the specific attributes that support their argument.

Additional quick notes:

On page 7, line 151, there are parentheses that are empty. Is there missing information or references that were omitted?

On page 16, line 345, there is a bolded note that says “(Error! Reference source not found.)”, which should be corrected.

6. PLOS authors have the option to publish the peer review history of their article (what does this mean?). If published, this will include your full peer review and any attached files.

Reviewer #1: No

Reviewer #2: No

Reviewer #3: No

---

## [Author Response · Author response to Decision Letter 0]

4 Nov 2022

PONE-D-22-00867 Inka Unku: Imperial or Provincial? State-Local Relations

Review Comments to the Author

Reply to reviewer:

Reviewer #1:

Query 1: Regarding the paper’s intent to “help us visualize the relationship between technological action and propagation of state power/ideology,” good observations are made regarding how local social agency (via technological aspects) is expressed. However, there are several areas where sentences or paragraph organization can be clearer. With regard to the parameters, for example, of “visible and low-visibility” attributes and the terms “technological” and “aesthetic,” sometimes the reader loses their way. The distinction of what is a technical attribute and what is an aesthetic attribute, for example, or how technology and style interface is not always precise and sections could be reviewed for more precision in word usage.

Reply: Agreed and fixed in the whole text. The term concepts of visible and low-visibility were expanded for clarification. This section of the paragraph now stands as (line 64-70): The concepts of metis and techne, or the relationship between local and State knowledge, can be materialized and understood through both high-visibility and low-visibility attributes, commonly used in the archaeological literature of enculturation. High-visibility attributes are intentional signaling features such as decorative/stylistic designs that denote standardized choices made by the government to transmit and create a sense of being part of larger corporate polity. Low-visibility attributes are technological features, such as raw materials, yarn types, or a particular textile structure, linked to group identity and community.

Query 2: Likewise, with content related to whether (local) adjustments to state standards are visible or not, there is a section (lines 471-498) where the reader loses sight of what the authors aim to do in terms of (as stated in line 87) bridging “the knowledge gap between visible and low-visibilty attributes.” I note in comments at end a text by Arnold and Espejo that is a good reference for how technology, structure, and motifs/aesthetics can be discussed in correspondence with one another. I don’t see it referenced in the bibliography and it would make sense to include it.

Reply: Agreed and fixed. Terms like “technological, structural, and aesthetic” were replaced by morphological, technological, and stylistic, throughout the entire text.

The entire paragraph was rephrased. Concepts from Arnold and Espejo (2013), and Hegmon (1995) were introduced in the text, which stands now as follows (line 91-97):

This section of the paragraph now stands as: To further analyze the unku, while following our theoretical perspective, we developed a methodological tool—based on parameters and attributes that consider morphological, technological, and stylistic features—that bridges the knowledge gap between high-visibility and low-visibility attributes, where morphology relates to textile composition; technology links social productive activities and their interactions with material culture [8]; and style here refers to “a form of non-verbal communication, through doing something in a certain way that conveys information about relative identity” [9:8]

Query 3: The tables provide relevant information re: standardization aspects in Inka unkus and the particular traits of the subject unku but there are a couple of things that might make the table 2 more digestible. One thing, for example, is that there could be color coding or something introduced in the Table 2 to highlight where CV unku’s features deviate from the Inka standard. This would be a helpful visual assist to the reader.

Reply: Agreed and fixed. A color coding was introduced in Table 2.

Throughout there are parts that bring up questions re: word choice or whether content is coming across clearly enough—I've made some notes below.

Section 1. Introduction

Query 4: From the Intro’s parag. 1 where authors note an attention to “technical-formal-aesthetic” elements but then later at Line 89 the reference is to “technological-structural-aesthetic” attributes. Should be consistent, formal is not equivalent to structural …or the equivalence authors are making should be clarified.

Reply: Agreed and fixed. As indicated in the answer to question 1, the attributes "technical-formal-aesthetic" and "technological-structural-aesthetic" were replaced by the terms "morphological, technological, and stylistic" throughout the text. 

Query 5: Lines 93-96—can be clearer—reference is made to “the unku” but it also seems reference is being made to unkus in general here, (e.g. “in some cases”) so which is the subject (one or all unkus available for study) of these sentences could be clearer.

Reply: Agreed and fixed. The paragraph stand now as follows (line 97-104): Our methodological tool is aimed to differentiate where in the chaîne opératoire the state canons differ from the local ones, thereby revealing the syncretic complexity of these iconic tunics (cf., Mary Frame’s [10] idea that hybridization in textile design was an effect of the technology used during the manufacturing process). It is designed for in-depth physical analysis to infer information about the identity of the intended wearer—as well as to understand the practical and functional purpose—of the garment [11]. This is based on the hypothesis that technical and/or stylistic differences in the unku might represent differences in the status of the wearer, which include the ruling elite, administrators, priests, and members of the military.

Section 2. Inka unku, imperial material imagery

Query 6: Lines 101-102 – re: “formal-functional, structural-aesthetic”-- the hyphenated relationships are a little confusing. Can it be more straightforward? E.g. technical (structural), aesthetic, functional, and socially recognized elements

Reply: Agreed and fixed. The entire sentence was rephrased as follows (line 125-127): Paramount homogenizing political, economic, and social organizations, such as the Inka State, established socially recognized morphological, technological, and stylistic elements that had to be considered by weavers.

Query 7: Line 105—last sentence seems out of place, perhaps belongs further up somewhere

Reply: Agreed and fixed. The sentence “Therefore, the unku were woven and distributed from Cusco across the entire territory under Inka control” was deleted.

Query 8: Lines 107-116 seem out of place here or unnecessary

Reply: Agreed and fixed. The entire paragraph was removed

Query 9: Line 120—re: not sure this is best word choice re: “disused”

Reply: Agreed and fixed. The word was replaced by used cloth.

Query 10: Line 122—sentence “During the early colonial period…” seems irrelevant?

Reply: Agreed and fixed. The sentence “During the early colonial period, they were worn for ecclesiastical feasts.” was deleted.

Query 11: Line 125—re: wording/word choice “’man-and-clothing’ relationship” is a little awkward.

Reply: Agreed and fixed. The entire sentence was replaced, and stands as follows (line 148-150): In Andean communities there is a relationship between clothing and people, insofar as the possession of certain types of clothing confers special social and cultural status to the wearer [22, 38].

Query 12: Lines 131-134: these first sentences don’t really correspond to the paragraph, could be removed

Reply: Agreed and fixed. The sentence was moved to the end of the previous paragraph, where it fits better, as it describes the different aspects surrounding the manufacture, use and disposal of unku.

Query 13: Line 136: suggest simplifying “consisted of six styles. This typology has been used to broaden…”

Reply: Agreed and fixed. The sentence was reworded, and now it reads as follows (line 183-185): To identify the State-imposed aesthetic in the production of the CV unku, we looked at the typologies proposed by John Rowe [7] and Ann Pollard Rowe [30] for Inka tapestry tunics, which consist of six styles. Ann Pollard Rowe expanded on John Rowe’s findings to develop a system that includes…

Query 14: Line 146: could re-phrase/check wording “developed politically”

Reply: Agreed and fixed. The wording was checked and the sentence stands as follows (line 196-199): Discontinuous warps and wefts were presumably woven for the elite [48]. The interlocking tapestry to create binary oppositions and the use of space were identified as a dominant textile features among Altiplano and highland social groups in the southern Andes [49:116].

Query 15: Line 153: can be clearer re: mention of one-way perspective here, since previous sentence refers to different perspectives. Can this sentence be re-phrased for clarity.

Reply: Agreed and fixed. The wording was checked and the sentence stands as follows (line 204-211): Most of the studies of unku noted above approach them from a technical and stylistic perspective primarily focused on identifying Inka influence on the provinces, which we consider a unidirectional perspective, concerned mainly with high-visibility attributes, or techne. In contrast, most of the 55 unku that were analyzed for the present study—from several museum collections from around the world (S1 Table)—were studied from the perspectives of both techne and metis.

Query 16: Line 155: Sentence re: “We believe that…” is very crammed with wording (of too many associations) and tries to do too much. Suggest breaking into separate sentences or re-phrasing for clarity. In other words, following the beginning of the next sentence, how are all these “points of convergence”?

Reply: Agreed and fixed. The sentence stands as follows (line 209-2015):

In this paper, we demonstrate that, when creating local unku, expert weavers followed the State canons, morphological, technological, and stylistic, to reproduce the emblematic symbols imbedded in these imperial garments (techne). Simultaneously, they integrated meaningful symbolic elements representative of local idiosyncrasies, traditions and experiential knowledge (metis). Consequently, the emblematic symbol of unku, instrumental for the expansion of the State, would have resulted from dynamic relationships between the Inka and the local community, in which expert weavers played a key role as agents of change.

Query 17: Line 159: Sentence “In other words” seems unnecessary here and disrupts rather than helps

Reply: Agreed and fixed. See query 15 (“In other words” was replaced with the term consequently).

Query 18: Lines 164-168: sentence “These creative processes…” can be revised for clarity—what is the takeaway here

Reply: Agreed and fixed. The sentence stands as follows (line 219-222):

These changes in the technical, formal, and aesthetic aspects of these textiles, resulting from the interaction between the State and the local community, which we believe represent the concepts of techne and metis, are part of what is known as provincial Inka [25, 29, 56, 57]. 

Query 19: Line 173: re: word choice-- “patterns” may not be the best word choice? State criteria? See note in 'Materials and Methods'

Reply: Agreed and fixed. That sentence was deleted and the paragraph now begins as follows (line 227-229):

In the Atacama region [58], local weavers introduced a series of hallmarks to the imperial attributes of the unku. For example,…

Query 20: Line 173: A couple of things are unclear here. Does the word “unku” here refer to the specific unku in this study or to unkus of this coastal region or to standard Inka unkus in general? Can clarify… the unku that is the subject of this study shares the camelid warp and weft criteria seen in Inka checkerboard unkus…but not all standardized state unkus (as referenced by Anne Rowe and John Rowe) have camelid weft and warp…some examples (esp. diamond waist) have cotton warp

Reply: Agreed and fixed. See Reply to query 18

Query 21: Line 179—if the previous sentence mentions the concealed gesture of multiple wefts, why does this sentence begin with “similarly” but then refer to the “simple weft” type that is recognized along Arica coast? It’s a little confusing.

Reply: Agreed and fixed. The sentence stands as follows (line 227-230):

In the Atacama region [58], local weavers introduced a series of hallmarks to the imperial attributes of the unku. For example, multiple weft were introduced, a traditional textile technique that became a concealed technical trait found in textiles during the Inka period [59-61]. On the coast of Arica, the continued use of single wefts has been recognized [60, 62].

Query 22: Lines 203-205—re: “…the search for… Inka state conventions in garments has been limited to analysis of their visible side”…unclear sentences in these 2 lines…sentence content can be clarified as implication is that no significant analysis has been made beyond just the visible

Reply: Agreed and fixed. The sentence stands as follows (line 249-255):

Whilst visibly reflecting the state’s aesthetic canons, provincial Inka styles were characterized by additional features or modifications, such as the use of different color combinations, geometric designs, or some form of iconography associated with the local landscape [67]. It appears, therefore, that the State posed no impediment to the application of local canons during the production process. To the contrary, it is likely the State allowed flexibility for agents to employ certain local practices that could be reproduced by hand while incorporating stylistic expressions of the State [14].

Query 23: Line 212—re: word choice “sceneries”

Reply: Agreed and fixed. The word was changed to “proposals”.

Section 3. State and provincial unku. Form, function, structure, and aesthetics

This subtitle was replaced by: State and provincial unku. Morphology, technology and style

Query 24: Line 237. Can clarify—ref. Rowe, in some standardized state unku examples, we do see cotton warp.

Reply: Agreed and fixed. We reviewed the entire paragraph, which stands as follows (line 271-282):

In the earliest phases of Andean history, vegetal fibers, such as Asclepias sp. and Cyperaceae sp. were commonly used to make fabrics [68-70]. Cotton does not appear on the coast and valleys of northern and central Peru until the Middle Archaic, becoming popular by the Late Preceramic period (5000 cal yr BP), largely supplanting the use of vegetal fiber in fabrics [62, 69, 71, 72]. Its use continued until the Late Horizon or Inka period. Camelid-fiber does not appear on the north coast until the Early Intermediate Period [73]. In contrast, in the valleys and coast of southern Peru and northern Chile, vegetal fiber was followed by camelid-fiber, and cotton was introduced during the early Formative (ca. 4000 cal yr BP) [68, 72, 74]. In Caleta Vitor, Martens and Cameron [74] confirmed that fabrics made of vegetal-fiber were common since the Early Archaic (ca. 9000-8000 cal yr BP). By Late Formative, however, this raw material drops off as camelid-fiber becomes the most common material. Similarly, the use of cotton was very limited, but became overwhelmingly dominant up to the Inka period [74].

Query 25: Line 256 – re: “special looms, hung on the wall,”….rather the suggestion from Rowe, Cobo, etc. is that the vertical looms were propped against a wall/ set upright against a wall…

Reply: Agreed and fixed. The sentence stands as follows (line 315-316):

Large textiles were woven using special looms, propped against a wall, with a vertical structure that consisted of cross bars, top and bottom, as well as heddles.

Query 26: Line 281—can be clarified, when you talk about the ahuaqui/awaki, the suggestion is that this is a reference to rhomboid/diamond motifs in general (according to the paragraph context) but the term is often used to refer to the area around the neck (the yoke) that may have stepped edges outlining a rhomboid… Or where is the citation for the ref. to Wari ahuaqui?

Reply: We agreed that the ahuaqui/awaki refers to a romboid/diamond motif/design. For this issue we added a reference by Berenguer 2013, page 330. The reference of this features related with Wari can be found in Campeny and Martel 2014, page 41, figure 8d.

Query 27: Line 289—unclear…what is meant by “in the case of the plain stripes…located around the arm slits”? What are the “plain stripes”? Is this a reference to woven stripes, as seen in the diamond cowl of the subject tunic, in which case the word would not be “applied to pieces” but “woven into the garment”…or is it reference to the embroidered striped cross-knit loop stitching (which would be “applied”) seen often on Inka tunic armholes and bottom edges?

Reply: Agreed and fixed. The sentence was rephrased as follows:

Unku with plain stripes located around the arm slits would have been worn by lower-level administrative functionaries[54, 86]. In the area of Arica, the mostly thin, plain-striped, natural-colored textiles have been interpreted as a simplified, decorative popularization of this Inka feature [22, 91, 92].

Query 28: Lines 319 and 322—unclear…but zigzag embroidery does not seem to be an element that lends the tunics more structure, which might be said of the cross-knit loop stitching at the hems or the figure-eight stitches on the side seams. Also, the suggestion in this paragraph is that all standardized unkus have zigzag stitching, but this is not the case…

Reply: Agreed and fixed. The statement “lends the tunics more structure” was deleted. Sentence was rephrased as follows (line 369-381):

The final stage in the production of an unku was to remove it from the loom and fold it in half along its vertical axis. The fold served to mark the area of the shoulders (from the wearer's point of view), and the neck slot was deliberately left open at the center of the tunic so it could easily be placed over the head. Both sides were sewn together to a sufficient height to allow the arms to be inserted. Tiwanaku, Wari, and Inka unku from the Central Andes feature high-density figure-eight stitching applied mainly in the lateral seams. This generated a double column that combined several colors, which gave an additional decorative element to the piece [7, 24]. The tunics also presents the zigzag in a way that combined the displacement and density of the fishbone stitch. Zigzag embroidery was typically applied near the bottom selvage of the tunic. In addition to the zigzag, the tunics show overcast stitch, known since Wari (AD 650–1000) [94], on the selvage edge. For Desrosiers [54], tapestry tunics from the coast are characterized by their reinforced longitudinal selvages, which can be observed in Tiwanaku, Wari, Inka, and probably colonial pieces. The tunics also retain their heading cords that produce the same effect as a reinforced selvage.

Section 4. Materials and Methods

Query 29: Regarding Sub-section “Standard Analytical Patterns / attributes/classification”- the first several paragraphs have a lot of repetition and the content could be streamlined. Suggest editing these paragraphs down to clarify main points. Everything could be summed up more or less in a few sentences instead of a few paragraphs: “Our method is based on semi-quantitative and qualitative analysis: we define a set of standard analytical patterns and their attributes on the Inka unku which, in turn, are understood as technical and aesthetic “hallmarks” of production. Framing our analysis within modes of production, we follow Splitstoser’s methodology, which recognizes…Our attempt to deconstruct process in the operation chain helps us understand….textile production norms.”

Question: why the word “patterns” to describe the analytical criteria? Not sure it’s the best word choice. “Patterns” is too resonant with how we would consider surface aspects of a textile (design/aesthetic, display/arrangement). Can another word be used to describe the considerations? “Criteria”? “Aspects”? “Elements”? “Parameters”?

Reply: Agreed and fixed. The following changes were made:

The term "patterns" was replaced by "parameters", and the first two paragraphs were rephrased as follows (line 461-468):

To define the standard analytical parameters and attributes associated with the production, use, repair, and discard of unku, we used a methodology developed by Splitstoser [62], which provides a practical framework for data collection based on a quantitative and qualitative analysis of textile attributes. Splitstoser’s method recognizes the various stages of the step-by-step chaîne opératoire of textile production [62]. The process spans raw material acquisition and preparation, yarn spinning and dyeing, loom choice and preparation, color planning, warping and weaving, off-loom finishing, use, reuse, repair, and disposal [50]. This analytical process allows the identification of the different uses of a textile throughout its life history [101].

Query 30: -Re: lines 432-433…what is the difference between “utilitarian features” and “functional particularities”?

Reply: Fixed. See Reply to query 29, where these differences were referred to Abal de Russo (2010). The term “utilitarian features” and “functional particularities”, however, were deleted. 

Query 31: -Re: line 432—“they were developed specifically for analyzing the unku..”—most of the “patterns” seem to refer to unku as a type rather than this particular unku…can this be clarified in this section?

This can also be clarified in the language used in Table 1, each section of which should be reviewed for language clarity and precision re: study parameter—e.g. sentence “we have divided the piece into separate parts,” can be clearer b/c don’t the authors mean “we have divided the Inka unku (as a type) into separate parts”? In other words, this is not about this specific piece but a general assessment of the type…but it’s not clear in the language.

Further issues regarding language clarity—for example, in “Attribute 7a. Diamonds or rhomboids,” the use of the word “structure” doesn’t seem appropriate here. The geometric shapes form the composition by outlining spatial areas. When the word “structure” is used, it confuses terms, since “structure” really connotes the interior/technical elements not the surface, design elements.

Reply: Agreed and fixed. We clarified that the clarification system refers any king of unku. The sentence stands know as follows (line 469-473): In Table 1 we list these nine analytical parameters and 26 corresponding attributes, arranged in numerical and alphabetical order. Table 2 presents the parameters and attributes used to study the CV unku. These terms are further defined in Supporting information S2 Table. These features are used to identify the technical and stylistic attributes that deviated from Imperial weaving canons and which are likely associated with the geographical origin of the textile.

See Supporting information S2 Table.

Attribute 5d. Spatial layout. The piece is observed and register analytically ("read") fully stretched from the weaver's point of view, from one end to the center, including the following analytical sections:…

Attribute 7a. Diamonds or rhomboids. Geometric shapes delimiting spatial zones with bilateral and/or quadripartite divisions.

Section 5. Results

Table 2

Query 32: Re: “Location of textile from weaver’s viewpoint” revise “Location on textile…”;

Reply: Agreed and fixed.

Query 33: Re: 2. Spinning, a. Yarn type, clarify: “Simple: regular/monochrome”

Reply: Agreed and fixed. See S2 Table, attribute 2a, yarn type.

Query 34: Re: 2. Spinning--the break up of the table here with parts c. through e. distributed differently—both on the horizontal and vertical axes-- and re: 3. Color Selection. Would it be useful to have this differently organized content put into a separate table to avoid having this irregularity in table organization?

Reply: Agreed and fixed. Table 2 was reorganized accordingly.

Section 6. Discussion

Query 35: Lines 457-458—these two sentences “More specifically” and “There must have” seem to disrupt the general gist and don’t serve the paragraph

Reply: Agreed and fixed. The paragraph was rephrased as follows (line 490-497): 

Results obtained from our methodological tool, provided us with a better understanding of the visual and technological features of the CV unku. We were able to characterize a series of visible traits strongly associated with the Inka State, suggesting this tunic would have been recognized as a state symbol of power and prestige at the time it was worn. If our analysis had focused solely on identifying and defining the visual attributes of Inka traits (techne), we would not have identified a series of traits unique to this piece and deriving, we propose, from local traditions (metis). The dialectical processes we identified between metis and techne must have been a source of tension between the skilled weavers and the technical and stylistic demands of the State.

Query 36: Lines 489-490—the suggestion that discrepancy in the alignment of the two parts of the large rhomboid is a “gesture of transgression” veers a bit from how the article frames local attributes in relation to state standards…should this idea of transgression be brought up sooner?

Reply: Agreed and fixed. The paragraph was rephrased as follows (line 527-531): 

Alternatively, it could be suggested that this “imperfection” came about due to an intentional decision on the part of the weavers to impart local/regional practices, or metis, to deviate from state norms. That being said, the evidence would not have been immediately visible, neither to the wearer nor the observer, given its position at shoulder height.

Query 37: Line 471 –498 –between these lines the arc of the argument varies from suggesting that local weavers were making [local] adjustments that were highly visible or also making adjustments that were not immediately visible. Content could be more carefully argued here. Re: the point that the unku carries local tradition and skill/experience (metis), was it meant to be seen or not? The takeaway gets muddled in these paragraphs.

Reply: Agreed and fixed. The paragraph was rephrased as follows (line 508-527): 

In one sense, the CV unku represents well-known, highly-visible Inka imperial standards (techne). This includes a long rectangular morphology and lack of sleeves. The measurements are within the established averages for width [between 72-79 cm; 6] and length [between 79-110 cm; 44]; see also Mary Frame [10]. Its interlocking tapestry technique is certainly of highland origin, as are its checkerboard design and “V”-shaped yoke, which forms a diamond when the piece is laid out fully. The diamond design may somehow reflect the Inka cosmovision of the world divided into four zones or suyus. On the other hand, the slit tapestry technique at the intersection of the pampa and diamond seems to refer to a coastal origin [10].

The CV unku presents several low-visibility structural and stylistic variants that we attribute to local/regional practices (metis), possibly of coastal origin. First, the textile was woven with cotton warp and camelid-fiber wefts. Because Imperial unku could have either camelid-fiber or cotton warps, this feature (cotton warps) is not diagnostic of an unku made on the coast origin; however, there is a strong association between cotton and coastal textiles. Although the CV unku was conceived as a single piece, it was made in two parts. This becomes visible where the sections converge, denoted by the heading cords that add structure to the shoulder area of the textile. At the shoulder seam, there is a small incongruity in the alternating colors of the diamond-enclosed stripes and pattern repetition, which appears to be the result of more than one weaver working together, as Rojas and Hoces de la Guardia [85] have proposed for other Inka textiles. Apart from the change of hands that may have led to these kinds of mismatches, they might also have resulted from a lack of rigor in the finishing process.

Query 38: Line 505—can clarify re: the half squares at the lateral selvages… the suggestion that half squares are a local anomaly but this does appear in many of the standardized BW unkus that John Rowe reviews.

Reply: Agreed and fixed. That statement was deleted, because we recognize that this was a mistake. The paragraph was rephrased as follows (line 545-546): 

On B side (from the wearer's point of view), two squares were woven half in very dark brown and half in dark brown.

Query 39: Lines 548-551—this content seems relevant as something to underscore earlier in the article as well.

Reply: Agreed and fixed. The following paragraph was added between lines 261 to 264 in the first section of the revised manuscript: Within this assemblage, the CV unku is the most eloquent form of material expression related to the Inka; therefore, we suggest that this textile was possibly locally produced in the context of a work commitment imposed under the mit'a system [14].

Query 40: Line 559--word choice, not “printed on the clothing” but rather introduced into/woven into /embedded into the woven structure of the cloth…

Reply: Agreed and fixed. The paragraph was rephrased as follows (line 593-598): The CV unku’s checkerboard design has been interpreted as a military symbol. The idea that the CV unku may have been worn by a local member of the State military is also supported by some of the associated archaeological materials (i.e., breastplate, and possibly the bow and arrows). Tentatively, we estimate that this function (military service) may have corresponded to another mit’a obligation incurred by the Caleta Vitor community. If this is the case, the person buried with this offering was of local origin.

Query 41: Line 564-565—points being made could be clearer, or seem to disrupt other content flow

Reply: Agreed and fixed. The paragraph was rephrased as follows (line 598-605): In Cusco, according to Murra [14], military garments were produced by women and given by the Inka government to men of military age, an endowment that represented a State reward. These tunics may have been produced by relatives of the soldiers. Similarly, we estimate that the State may have gifted the locally-produced CV unku to this military figure as a prestige object befitting his position. In this process of giving and receiving, the State enacted the redistribution system of mit’a-produced objects with the community of Caleta Vitor. The action of giving helped bring negotiations and agreements to a successful conclusion, paving the way for integration and fostering collaboration between the community and the State.

Query 42: *Throughout double check for spelling, punctuation--, eg p. 28 table 2 heading– typo in “application”; sometimes “state” is capitalized, sometimes not…

Reply: Agreed and fixed. The word introduction was corrected as mentioned. State term was checked too. State with capital letter was used when it refers to the State as a subject. State with lower case was used when term acted as adjective (state patters). There are other contexts for this term like “state of preservation”, or used as verb (e.g., Arnold also states). Other words were fixed too.

Query 43: Re: recommend including/using these additional sources:

Mary Frame, “Chuquibamba: A Highland Textile Style,” Textile Museum Journal, 1997-1998, Vols. 36, Washington DC. Frame’s essay regarding provincial Inka tunics may be of interest as she discusses various aspects of textile production from the Chuquibamba textile tradition and contextualizes them against Inka standards to a degree. Also, at parts, she considers how fabric-making technology informs design in textiles (contrast to this article’s interest in pointing to chains of operation at the local level as being influential over design aspects).

Reply: Mary Frame´s study was integrated in several section of the manuscript (see lines 99, 200, 222, 511). 

Query 44: Denise Arnold and Elvira Espejo, El Textil Tridimensional: La Naturaleza del Tejido Como Objeto y Como Sujeto, Edition: Serie Informes de Investigación, II, No. 8, Fundación Xavier Albó and Instituto de Lengua y Cultura Aymara, La Paz, 2013.

Reply: The study by Denise Arnold and Elvira Espejo was also integrated of the manuscript (see line 95) and in the Supporting Information S2 Table (Parameter 3).

Query 45: See pp. 118-130 of the book--This article’s bibliography references Denise Arnold’s Textil y Tributo but I would also point to this text which discusses weaving tools and practices within the notion of chaine operatoire, particularly, that may be relevant to this article. For example, although in an ethnographic context and within an Aymara context, the authors refer to what goes into the color selection process within the operation chain fairly in-depth, referencing musa waraña. This may be useful to this article for elaborating on some points re: how color choices are made/were made perhaps in manner applicable to the question of operation chain re: color and Inka context, etc.

See also pp.183- 184 in same book. These paragraphs tease out question of how to think about style in relation to technology and may be useful, would be relevant to cite.

Reply: Agreed and fixed. These details were also integrated in the Supporting Information (see S2 Table, Parameter 3).

Query 46: Also, the bibliography points to the text “An Inka Unku from Caleta Vitor Bay, Northern Chile,” written by some of this article’s same authors, with citation/note in Attribute 5b. That text’s attention to the same unku and question of Inka investment/stamp on the CV unku garment seems pertinent to this article’s inquiry and probably should be given further mention, e.g. stating how this extends/expands/or deviates from that study.

Reply: Regarding the article by Martens et al. 2021, which includes some of the co-authors of this manuscript, was aimed to confirm that the unku from Caleta Vitor was an authentic Inka imperial garment, comparing its technical and stylistic characteristics of 36 examples of unku, recorded in museums of Europe and the United States of America. Martens et al. 2021 concluded that the unku from Caleta Vitor could be classified as a new type, and added a new perspective to "the complex relationship between the inhabitants of this province and the Inka State". It was also noted that "more research is needed on late period interactions in northern Chile and the artifact assemblage typically associated with the Inka." In the process of expanding the study of this particular unku we realized that the state parameters were accompanied by features that did not conform to the stylistic and technological typologies previously established by other authors. This motivated us to develop a methodological procedure to distinguish more precisely in which part of the operative chain the state canons differ from the local ones. From this observation and based on the literature, we began to put forward the ideas and arguments of this new manuscript.

Part of this explanation was added to main text of the manuscript (see lines 84-90).

 

Reviewer #2: 

The authors present a fascinating and potentially very important argument: that Inka unku, paradigmatic symbols of imperial power, are made locally in certain cases. This would be a welcome addition to this area of research. That said, the article would be strengthened by clarifying and quantifying key aspects of the argument that at the moment are either unsupported or clouded by extraneous detail.

Query 47: For example, at the outset the authors argue that the Inka "imposed technical and aesthetic canons" for the production of the tunics. How did they do this? It would be good to note the data supporting this statement. Furthermore, how does this then play into this article's argument that this tunic was outside (?) of this control? Or, if not outside of this control, how were these canons shared/imposed?

Reply: The following paragraph was added to support our argument that the State needed to find ways to communicate/transmit and ultimately impose its stylistic parameters, within a dynamic process that left room for local innovations:

The State imposed a series of labor obligations and commitments through mit'a (asymmetric reciprocal labor system) using different coercive measures including political, economic, military and even matrimonial agreements with local communities. In this system of reciprocal relations, the State provided the necessary resources (raw materials, tools, architectural facilities, etc.), for the performance of assigned tasks [14, 39]. The mit'a agreements also implied great acts of generosity on the part of the State as another way of committing local communities to participate in this reciprocal process [14]. In this context of generosity, we believe that the State was also able to transmit the stylistic, architectural, morphological, and technological patterns that were expected (?) to be applied in local (?) production, all of which involved acculturation processes [40]. This would explain, in part, the standardization of material and immaterial culture that characterized the Inka State throughout the Andes [41]. 

Standardization, such as that found in the unku, is likely related to the need of the State to reproduce its symbols of power. Therefore, garments had to adhere as strictly as possible to the standard parameters that gave these pieces an eminently imperial character; however, it has also been pointed out that one of the political strategies of the Inka State was to coordinate the diversity of the peoples, with their different cultures, languages, and traditions, without trying to homogenize them completely. The Inka promoted diversity and allowed local groups to produce material culture and conduct local ritual behavior in ways that represented their ethnic identity [14, 42]. In short, the relationship between the state and local communities was a dialectical process that varied from region to region and village to village, depending on the pressure exerted by the state and the receptivity of local groups [40, 43]. In our opinion, these historical contexts would have generated the conditions for local weavers to introduce morphological, technological, and stylistic features with low visibility, attributes which did not radically transform the imperial character of these iconic objects that represented of the State.

Query 48: Most importantly, however, the crux of the argument lies in the presumed fact that the tunic was made locally. This is an essential piece of the argument and it needs to be presented in a clearer, more forceful way. The article will be strengthened by cutting back on jargon and presenting the data in a straightforward way. At the moment it reads more like a discursive thesis than a scientific journal piece.

Reply: The characteristics that distinguish this piece from being of local manufacture were clarified in the answer to query 37 of Reviewer #1; all of which was introduced in the main text (lines 529-541). As for the “jargon”, if it refers to the technical analysis of the textile, it is necessary to point out that the theoretical-methodological procedure presented in this manuscript is key to substantiate the ideas proposed here. In addition, following the suggestions of reviewer #1's question 34, who seems to be comfortably with the technical terms, the contents of Tables 1 and 2 were reviewed in detail. If this is not the case, it is difficult to imagine what jargon the reviewer is referring to specifically. It would have been ideal to have comments as precise as those of Reviewer #1.

Query 49: I encourage the authors to tighten up their manuscript and re-submit as the research is of much merit.

Reply: The whole manuscript was revised, edited and tighten, following comments of the three reviewers. 

 

Reviewer #3:

Query 50: 

a) On page 5, the authors write that their objective is “to bridge the knowledge gap between visible versus low-visibility attributes. To this end, we created and defined standard analytical patterns and their technological structural-aesthetic attributes to analyze the unku. This theoretical-methodological tool also serves to evaluate the possible existence, or not, of technical and/or aesthetic differences in the unku, depending on the status of the wearer.” This objective, and the relationship between it and the specific analytical approach taken to characterize the different variables of the unku, remains deeply confusing to me. First, I do not know what a “theoretical-methodological tool” is; theory refers to a conceptual framework used to interpret patterns in the archaeological record that asserts specific relationships between variables, and that can be objectively tested against predicted patterns in the record. To the extent that a theoretical framework is offered, this draft perhaps predicts variability between textiles’ attributes that can be attributed to, or used to identify, local weavers’ agency.

Reply: Agreed and fixed. We recognize the confusion created by the terms theoretical-methodological tool. 

The term theory is used only to refer to the following concepts: metis and techne, mit'a, low-visibility and high-visibility that support the central theoretical approach of this manuscript, which proposes that the weavers, mainly local, although they had to and did follow the morphological-technological-stylistic parameters of the Inka State, sought ways to introduce features that gave certain particularities to the tunics without altering the iconographic image of the State. Consequently, following this comment, the theoretical and methodological terms were completely separated in the main text.

b) Methodology refers to a specific set of sampling and data collection techniques that can be used to collect the data needed to objectively test specific hypotheses generated by the theoretical framework. If the analytical form in Table 2 is what the authors as referring to by the phrase “theoretical-methodological tool”, alternatively referred to on page 20 as “a theoretical practical framework for data collection”, it needs to be explained how this particular spreadsheet of attributes derives from the theoretical approach they advocate for: how, for example, the assertion that local weavers used specific technical or aesthetic differences to indicate the status of the wearer.

Reply: Agreed and fixed. In lines 97 to 108, we added the following paragraph to introduce the purpose and characteristics of the methodology: Our methodological tool is aimed to differentiate where in the chaîne opératoire the state canons differ from the local ones, thereby revealing the syncretic complexity of these iconic tunics (cf., Mary Frame’s [10] idea that hybridization in textile design was an effect of the technology used during the manufacturing process). It is designed for in-depth physical analysis to infer information about the identity of the intended wearer—as well as to understand the practical and functional purpose—of the garment [11]. This is based on the hypothesis that technical and/or stylistic differences in the unku might represent differences in the status of the wearer, which include the ruling elite, administrators, priests, and members of the military. Because each unku was woven for one of these characters, with a specific role and status, were weavers allowed (or took) a significant control over the way they conformed to (or deviated from) imperial versus local weaving styles and parameters? Furthermore, in some cases, unku acquired additional elements and underwent modifications for reuse and when being discarded.

For the second concern of the query, the phrase “a theoretical practical framework for data collection”, was fixed as follows: “a practical framework for data collection” (see line 463)

c) It is also unclear why a checkerboard unka (male tunic) buried with two young women in a provincial burial is appropriate or sufficient data to test this hypothesis.

Reply: It is necessary to say that the women of the Inka sanctuary of the coast of Iquique (Cerro Esmeralda) do not have a checkerboard unku. Abal de Russo 2010 [11] mentions a checkerboard unku associated with a girl sacrificed at the Chuscha volcano. The author wonders if the girl was specially dressed in male clothing for that ritual [11:261]. At Aconcagua volcano, a miniature of a checkerboard unku associated with a male statuette belonged to a child of the same sex. Regarding the evidence from the Chuscha volcano, since the excavations were made by amateurs, the circumstances of the findings were not well documented. The author [11:260, 394] notes, moreover, that these garments are typically part of male clothing. In sum, the evidence is rather weak to associate this clothing with women.

Query 51: On page 5, the authors assert that “he unku were woven and distributed from Cusco across the entire territory under Inka control.” This is directly contradicted by the statement on pages 8-9 (emphasis added) that “local weavers introduced their own hallmark in the form of multiple wefts, a traditional textile technique that became a concealed technical gesture used in textiles during the Inka period [47-49]. Similarly, continual use of the simple weft has been recognized along the Arica coast.”

Reply: Agreed and fixed. Following Reviewer #1 query 7, the sentence “Therefore, the unku were woven and distributed from Cusco across the entire territory under Inka control” was deleted as it is “out of place”. This means that we maintain the statement of pages 8, and lines 157-158.

Query 52: 

a) Indeed, I believe a better statement of the paper’s main contribution appears on page 10, where the authors write: “we have developed a decoding tool for the unku with the aim of distinguishing state from local hallmarks, thereby revealing the syncretic complexity of these iconic tunics.”

Reply: Agreed and fixed. The statemen “we have developed a decoding tool…” was integrated in the abstract (lines 29-30), as well as in lines 97-108, which is part of the introduction of the manuscript (see Reply to query 50). 

b) If other scholars have assumed that unku were centrally produced in Cuzco and distributed throughout the empire, then your study provides additional support to Williams et al, and others’ work demonstrating local production of imperial-style ceramics, but extended to weaving production. Consider reframing the article around this narrower, but clearer objective.

Reply:

Previous studies on the unku have focused their interest on specifying the technological and stylistic parameters that identify these tunics as imperial, so they have not been concerned with identifying whether production occurred in Cusco or in the provinces. In the original version of our manuscript we indicated, in a confusing way, that the unku were produced in Cusco. That sentence was deleted following comments from reviewers 1 and 3 (see response to questions 7 and 51, respectively). Our study follows in part the argument of Williams et al. (2016), who point out that local ceramic artisans tried to emulate, as much as possible, the imperial styles in their visible aspects, and in that process would have added consciously or unconsciously certain traditional procedures, such as the use of local raw materials that differed from those used in the state centers of ceramic production (ollero villages) in the Andean zone. Their ceramological analysis demonstrated that although the raw materials were different, the artisans tried to give the vessels an imperial appearance. Hughes [27] points out a similar situation for the production of chuspas.

Our study, however, goes a step further than that indicated by Williams et al. and Hughes, insofar as to identify the syncretism that integrated, on the one hand, the imperial parameters, obviously more visible and, on the other hand, certain local parameters, less visible, by proposing a methodology aimed at identifying at what point in the long process of production, use and discard of the tunics, the local contributions were introduced and what were their characteristics.

In short, given that we have stressed this objective, highlighted not only by this reviewer, throughout the entire manuscript, the suggestion of "reframing the article around this narrower, but clearer objective", it is accomplished. Regarding the conclusion of Williams et al., and other authors, we rephrased that paragraph to make clearer how this key issue is connected with our arguments (see lines 234-248).

Query 53: I also believe that the very long list of complete attribute descriptions given in Table 1 can be selectively condensed. I would much rather, for example, have the full list – along with the form given in Table 2 – put on a website somewhere. The space could then be devoted to explaining with 2-3 specific examples how local weavers’ agency would produce different values in specific attributes than centrally-controlled or Cusqueño weavers’ textiles. In other words, the article could use a much fuller and more specific, detailed explanation of how specific attributes in the form are useful for identifying local weavers’ choices or hallmarks. Using the form, or only a select subset of attributes from the form, to show distinct differences between the Caleta Vitor unku and one or more unku that were clearly made by Cusqueño weavers, would provide a stronger argument for the form’s utility in testing hypotheses about state vs local differences in the chain of operations. The authors begin to do so on pages 34-36 in the discussion section, but I would like to see a much more detailed dive into the specific attributes that support their argument.

Reply:

Table 1 was moved to Supporting Information (S2 Table), but it was replaced in the main text by a summary listing the 9 analytical parameters and their 26 attributes. 

Table 2, however, was maintained in the main text, since it shows in detail the morphological, technological, and stylistic parameter of imperial and local origin. To make it clearer a column was added to this Table, indicating for each attribute whether it represents an imperial or local feature. This addition makes it possible to visualize the elements that are considered local based on data from previous studies and our own experience. 

The suggestion to apply the analytical tool to "one or more unku that were clearly made by Cusqueño weavers" is beyond the scope of this study, but we have set it as a task for a next step in this research.

Additional quick notes:

Query 54: On page 7, line 151, there are parentheses that are empty. Is there missing information or references that were omitted?

Reply: Fixed.

Query 55: On page 16, line 345, there is a bolded note that says “(Error! Reference source not found.)”, which should be corrected.

Reply: Fixed.

---

## [Decision Letter · Decision Letter 1]

14 Dec 2022

PONE-D-22-00867R1Inka Unku: Imperial or Provincial? State-Local RelationsPLOS ONE

Dear Dr. Correa-Lau,

Thank you for submitting your manuscript to PLOS ONE. After careful consideration, we feel that it has merit but does not fully meet PLOS ONE’s publication criteria as it currently stands. Therefore, we invite you to submit a revised version of the manuscript that addresses the points raised during the review process. Both reviewers are very positive about your revised manuscript. Reviewer 1 identifies several issues that should be addressed when making your final revisions. Please be sure to do a thorough proofread of the manuscript before submitting it.

We look forward to receiving your revised manuscript.

Kind regards,

John P. Hart, Ph.D.

Academic Editor

PLOS ONE

Journal Requirements:

Reviewers' comments:

Reviewer's Responses to Questions

**Comments to the Author**

1. If the authors have adequately addressed your comments raised in a previous round of review and you feel that this manuscript is now acceptable for publication, you may indicate that here to bypass the “Comments to the Author” section, enter your conflict of interest statement in the “Confidential to Editor” section, and submit your "Accept" recommendation.

Reviewer #1: (No Response)

Reviewer #3: All comments have been addressed

2. Is the manuscript technically sound, and do the data support the conclusions?

Reviewer #1: Yes

Reviewer #3: Yes

3. Has the statistical analysis been performed appropriately and rigorously? 

Reviewer #1: N/A

Reviewer #3: N/A

4. Have the authors made all data underlying the findings in their manuscript fully available?

Reviewer #1: Yes

Reviewer #3: Yes

5. Is the manuscript presented in an intelligible fashion and written in standard English?

Reviewer #1: No

Reviewer #3: Yes

6. Review Comments to the Author

Reviewer #1: I recommend minor revisions before this is completely ready for publication. It would be very worthwhile to do a full copyedit review as there are a few places that need to be tended to for grammar, punctuation, sentence structure, or something along those lines. See, for example, lines 91-97; 105-108; 156-158; line 167; line 182; lines 224-226; 228-230; 244-245; 289-290; 302-303 (here also some of the content re: yarn spin could be put in a footnote, doesn’t all seem necessary here); lines 568-570. These are parts where I caught some kind of error or oversight but a professional copyeditor would be most useful.

General comment: suggest that words like “state” and “imperial” be lower case across the board. There are still some inconsistencies in capitalization of these here and there and it would be cleaner to just lower case these words throughout, even re: “Inka state.”

Other general comments/notes:

Line 53—what is the “importance of the relationship btwn state and local knowledge” what is the “dialectical arena in which state ideologies could be negotiated”

Line 79: suggest removing first clause, re: “we recognize style …tool of social action” and beginning sentence with “The State-produced textiles” b/c what ‘style’ encompasses is vague here and subsequent sentences elaborate on a variety of things besides style that become ‘tools of social action’…technical aspects, manufacture/process, etc..

Lines 109-121--these do not seem necessary to the text and maybe muddle the direction the discussion is headed in. If anything, the quoted text (re: “technology, as embodied…”) might be preserved but moved to a different section, prob. somewhere in previous pages;

Line 168…sentence “This would explain,..” is confusing, would delete;

Line 196 “discontinuous warps and wefts …woven for the elite” is a confusing phrase;

Line 469…the different ‘stages’? vs. ‘uses’ of a textile’s life history?;

Lines 546-550—this is so interesting about the half dark brown/half very dark brown squares, but can there be elaboration here as to why this would be local hallmark versus other explanation? Do we see examples of this kind of chromatic duality locally, that can be pointed to here for support? Also, the statement is that these half-squares are not conspicuous but they’re located or seem to be located toward the center of the textile panel, which would seem to be more conspicuous than not (?)

Line 566, a little unclear-- what is the ‘new’ role that the mending presumably highlights, apart from the original role the piece would have had before the mending…

Line 621—in the context of broader Andean cultural practices, another way of saying this might be the ‘tradition of giving and receiving ‘coopted’ by the state” rather than “‘established’ under the state”

Line 626-628 and ‘syncretic conditions’ in textiles that cannot be found or repeated in textiles—but in other parts of the paper the comment seems to be that ceramics can also display hallmarks of local tradition along with state aspects (e.g. pp. 11, 27)

Line 632—but what comes up in the content is that not all are ‘masked hallmarks’...that should be included here for clarity/nuance

Reviewer #3: Thank you for your detailed and careful attention to the revisions. It is now a well-written, clearly organized, and tightly focused argument. I look forward to seeing the final product in print. Please note though the phrase "Error! Reference source not found" that appears in line 401 (page 18). That is the only one I found.

7. PLOS authors have the option to publish the peer review history of their article (what does this mean?). If published, this will include your full peer review and any attached files.

Reviewer #1: No

Reviewer #3: No

---

## [Author Response · Author response to Decision Letter 1]

30 Dec 2022

PONE-D-22-00867/R1: Inka Unku: Imperial or Provincial? State-Local Relations.

Review Comments to the Author

Reply to Reviewer #1: 

Query 1: I recommend minor revisions before this is completely ready for publication. It would be very worthwhile to do a full copyedit review as there are a few places that need to be tended to for grammar, punctuation, sentence structure, or something along those lines. See, for example, lines 91-97; 105-108; 156-158; line 167; line 182; lines 224-226; 228-230; 244-245; 289-290; 302-303 (here also some of the content re: yarn spin could be put in a footnote, doesn’t all seem necessary here); lines 568-570. These are parts where I caught some kind of error or oversight but a professional copyeditor would be most useful.

Reply: Agree and fixed. The text was reviewed and corrected in its entirety by a native English speaker 

Query 2: General comment: suggest that words like “state” and “imperial” be lower case across the board. There are still some inconsistencies in capitalization of these here and there and it would be cleaner to just lower case these words throughout, even re: “Inka state.”

Reply: Partially agreed and fixed. The Associated Press (AP) and the Chicago Manual of Style, indicate that the word "state" is capitalized when referring to a government agency. In all other cases they prefer lower case. Therefore, we have opted for "Inka State" (and Inka Empire) but "state" (lowercase) for all other cases. Besides, different articles (mostly Plos One) speak of Inka/ Inca State. We maintained the Inka term capitalized because is a proper name.

Query 3: Line 53—what is the “importance of the relationship btwn state and local knowledge” what is the “dialectical arena in which state ideologies could be negotiated”

Reply: Fixed. A long sentence has been added to deepen the conceptual terms pointed out, which stands as follows (see lines 53-62): In this regard, we address the importance of the relationship between state and local knowledge, which brought agents into a dialectical arena in which state ideologies could be negotiated. The importance lies in the fact that, despite the scale of the social and military structure of the Inka State, its expansive, economic and ideological success depended to a large extent on the agreements reached, peacefully or belligerently, with the local communities of the provinces integrated into the empire. Among the dialectical scenarios was the textile industry, in whose production the technical and idiosyncratic canons of the state were negotiated with those of the communities themselves. In this context, the unku, as symbols of imperial power, are key elements for materially visualizing the dialectical relationship between the state and the local communities.

For the analysis of this dialectical arena, we use the concepts of metis and techne [1] to facilitate an understanding of the dialectical relationship between these agents.

Query 4: Line 79: suggest removing first clause, re: “we recognize style …tool of social action” and beginning sentence with “The State-produced textiles” b/c what ‘style’ encompasses is vague here and subsequent sentences elaborate on a variety of things besides style that become ‘tools of social action’…technical aspects, manufacture/process, etc..

Reply: Agree and fixed. The paragraph was rewritten and stands as follows (see lines 86-90): The technological styles and symbolic structures of the textiles produced for the state constituted an active tool of social action [5] insofar as their production depended on the weavers of the local communities who had to follow the state model. In this active role of perpetuating state canons, local weavers had the insight to introduce subtle technical and stylistic changes throughout Tawantinsuyu.

Query 5: Lines 109-121--these do not seem necessary to the text and maybe muddle the direction the discussion is headed in. If anything, the quoted text (re: “technology, as embodied…”) might be preserved but moved to a different section, prob. somewhere in previous pages;

Reply: Agree and fixed. Part of the text of this paragraph was moved to a previous page, which stands as follows (see lines 101-106): Technology links social productive activities and their interactions with material culture [8], which means that “technology, as embodied material practice, is a socially charged and materially grounded arena in which agents express and negotiate social relationships, establish and express value systems, and give meaning to the object world” [9:162]. Technology, then, is recognized as a meaningful social activity, driven through agency processes [10].

Query 6: Line 168…sentence “This would explain,..” is confusing, would delete;

Reply: Agree and fixed. The full sentence was deleted.

Query 7: Line 196 “discontinuous warps and wefts …woven for the elite” is a confusing phrase;

Reply: Agree and fixed. The sentence was rephrased as follows (lines 203-206): Discontinuous warps and wefts were presumably woven “to signal the office of the person who wore that class of garments” [48:230]. Consequently, Dransart, has suggested that “it is possible to see different levels of social articulation in these garments” [48:230].

Query 8: Line 469…the different ‘stages’? vs. ‘uses’ of a textile’s life history?;

Reply: Agree and fixed. The sentence was rephrased as follows (lines 475-476): This analytical process makes it possible to identify the different stages of the textile's life history [101].

Query 9: Lines 546-550—this is so interesting about the half dark brown/half very dark brown squares, but can there be elaboration here as to why this would be local hallmark versus other explanation? Do we see examples of this kind of chromatic duality locally, that can be pointed to here for support? Also, the statement is that these half-squares are not conspicuous but they’re located or seem to be located toward the center of the textile panel, which would seem to be more conspicuous than not (?)

Reply: We particularly appreciate this comment, and we, added the following phrase to clarify this issue (lines 554-566): This characteristic of two squares with chromatic duality in the brown range is exemplified on a large scale (i.e., the entire surface of the textile) in two unku defined as provincial in the collection of the Ethnological Museum of Berlin (VA 16630 and VA 4576) [105]. Another case, but with curved figures, is observed with a duality of blue range (VA 62696). The CV unku, a piece that integrates imperial and provincial morphologies, technologies, and styles, is a perfect example of syncretism, which transforms it into a local product. The detail of the chromatic duality in a single square of a checkered surface has not been noted in any other unku of the BW type, with which this piece would be associated. Therefore, we suggest investigating this feature as an element of the creation of local weavers who, following the state templates, introduced distinctive hallmarks. As for the prominence of this feature, although it is located on the front flange of the garment, due to slight tonal differences, it is not readable. Similarly, it occurs with the intersection of the stripes at shoulder height, which are in less conspicuous positions. We identified both features as having low-visibility.

Query 10: Line 566, a little unclear-- what is the ‘new’ role that the mending presumably highlights, apart from the original role the piece would have had before the mending…

Reply: The whole paragraph was rewritten as follows (lines 577-587): Repairs to the piece were made by intricately and delicately emulating the original technique. As such, they are not immediately apparent and would have been barely perceptible to observers. For this careful repair work, the mender used composite yarns of different colors to those of the original textile, thus disclosing the intention to modify the conservation of the piece and unveil his hallmark as a mender. The repairs were intended to prolong the lifespan of the piece and thus give way to the second usage (reuse) of its life history. The CV unku was a singular piece in which, at first glance, state and provincial canons are identified, which thus made it an object valued locally and representative of a social, cultural, geographical, and temporal context. In this sense, this piece was transformed into an inheritable and desirable object given its ideological and political charge, which was to be maintained in perpetuity through reparations, thus endorsing the position of the character or family lineage in which the piece fell, to the extent that it continued to be part of the structure of the political relationship between the state and this local community.

Query 11: Line 621—in the context of broader Andean cultural practices, another way of saying this might be the ‘tradition of giving and receiving ‘coopted’ by the state” rather than “‘established’ under the state”

Reply: Agree and fixed.

Query 12: Line 626-628 and ‘syncretic conditions’ in textiles that cannot be found or repeated in textiles—but in other parts of the paper the comment seems to be that ceramics can also display hallmarks of local tradition along with state aspects (e.g. pp. 11, 27)

Reply: Agree and fixed. The sentence was rephrased as follows (lines 646-649): This syncretic condition of the textile could also be repeated in other goods produced by and for the state. For example, despite the rigid production processes of ceramics, artisans managed to introduce technical and aesthetic variations that resulted in a diversity of vessels defined as local or provincial Inka.

Query 13: Line 632—but what comes up in the content is that not all are ‘masked hallmarks’...that should be included here for clarity/nuance

Reply: Agree and fixed. The sentence was rephrased as follows (lines 652-654): The tension embodied by weavers in having to explicitly reproduce state patterns, and mask or subtly unveil their local hallmarks are well represented and expressed in this unku.

Reply to Reviewer #3:

Query 1: Thank you for your detailed and careful attention to the revisions. It is now a well-written, clearly organized, and tightly focused argument. I look forward to seeing the final product in print. Please note though the phrase "Error! Reference source not found" that appears in line 401 (page 18). That is the only one I found.

Reply: Agree and fixed.

---

## [Editor Report · Decision Letter 2]

3 Jan 2023

Inka Unku: Imperial or Provincial? State-Local Relations

PONE-D-22-00867R2

Dear Dr. Correa-Lau,

We’re pleased to inform you that your manuscript has been judged scientifically suitable for publication and will be formally accepted for publication once it meets all outstanding technical requirements.

Kind regards,

John P. Hart, Ph.D.

Academic Editor

PLOS ONE
---

## [Editor Report · Acceptance letter]

11 Jan 2023

PONE-D-22-00867R2 

Inka *Unku*: Imperial or Provincial? State-Local Relations 

Dear Dr. Correa-Lau:

I'm pleased to inform you that your manuscript has been deemed suitable for publication in PLOS ONE. Congratulations! Your manuscript is now with our production department. 

Kind regards, 

on behalf of

Dr. John P. Hart 

Academic Editor

PLOS ONE